# Wait, Wait, Wait...
# Why Do Reasoning Models Loop?

**Charilaos Pipis** [* 1] **Shivam Garg** [* 2] **Vasilis Kontonis** [2]
**Vaishnavi Shrivastava** [2] **Akshay Krishnamurthy** [2] **Dimitris Papailiopoulos** [2 3]

## Abstract

Reasoning models (e.g., DeepSeek-R1) generate long chains of thought to solve harder problems, but they often loop, repeating the same text at low temperatures or with greedy decoding. We study why this happens and what role temperature plays. With open reasoning models, we find that looping is common at low temperature. Larger models tend to loop less, and distilled students loop significantly even when their teachers rarely do. This points to mismatches between the training distribution and the learned model, which we refer to as errors in learning, as a key cause. To understand how such errors cause loops, we introduce a synthetic graph reasoning task and demonstrate two mechanisms. First, risk aversion caused by hardness of learning: when the correct progress-making action is hard to learn but an easy cyclic action is available, the model puts relatively more probability on the cyclic action and gets stuck. Second, even when there is no hardness, Transformers show an inductive bias toward temporally correlated errors, so the same few actions keep being chosen and loops appear. Higher temperature reduces looping by promoting exploration, but it does not fix the errors in learning, so generations remain much longer than necessary at high temperature; in this sense, temperature is a stopgap rather than a holistic solution. We end with a discussion of training-time interventions aimed at directly reducing errors in learning.

[1]MIT [2]Microsoft Research [3]University of Wisconsin-Madison. Correspondence to: Charilaos Pipis <chpipis@mit.edu>, Shivam Garg <shigarg@microsoft.com>.

*Proceedings of the $43^{rd}$ International Conference on Machine Learning*, Seoul, South Korea. PMLR 306, 2026. Copyright 2026 by the author(s).

## 1. Introduction

Reasoning models (Jaech et al., 2024; DeepSeek-AI et al., 2025; Abdin et al., 2025; Guha et al., 2025) use extra inference time compute, generating long chains of thought, to solve harder problems. This has opened a complementary scaling axis of inference-time compute, alongside training compute, resulting in striking gains on challenging tasks such as competitive math and coding. Yet these models often get stuck in loops: endlessly repeating the same text in their chain of thought, especially under greedy decoding and low temperatures (see Section 2.1 for an example). As a result, most model providers recommend running them at a sufficiently high temperature to avoid looping (e.g., see the Hugging Face pages for DeepSeek-R1 and QwQ-32B).

This raises several questions: why do these models loop, and how does temperature help? In particular, does temperature address the root cause or mostly act as a stopgap? Ideally, temperature would be a knob we can use to control how much exploration a chain of thought performs, rather than something we must turn up just to avoid looping. More fundamentally, is randomness a necessary resource for good reasoning models? This is reminiscent of classical questions in algorithms about whether randomized algorithms are more powerful than deterministic ones (Motwani & Raghavan, 2013; Vadhan, 2012). In this work, we take a step towards understanding these questions. Our contributions are as follows.

**Observations with open reasoning models (Section 2).** We evaluate several open reasoning models (e.g., DeepSeek–distilled Qwen, Openthinker-3, Phi-4 reasoning) for looping on problems from the American Invitational Mathematics Examination (AIME), a high-school math contest. We make several observations: (i) all models loop at low temperatures; (ii) within a family, smaller models loop more; (iii) for models trained via distillation, students loop far more than their teachers; and (iv) for most models, harder AIME problems elicit more looping. These observations point to imperfect learning, that is, systematic *errors in learning* of the training distribution, as a key cause. If a student perfectly learned the teacher, then the amount of looping of the student could not be significantly higher than

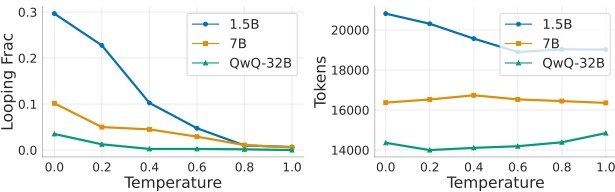

*Figure 1.* **Looping and response length for OpenThinker models.** Looping fraction and response length for OpenThinker3-1.5B/7B (students) and QwQ-32B (teacher), averaged over AIME 2024/2025. For each (problem, model, temperature) we sample 20 chains; a response is looping if some 30-gram appears $\geq$ 20 times. (Left) Students loop much more than the teacher at low temperature. (Right) While increasing temperature largely removes looping, the student models still produce substantially longer responses than the teacher at high temperatures.

that of the teacher. For instance (Fig. 1), how can it be that OpenThinker3-1.5B loops in 30% of its responses with greedy decoding, while its teacher barely loops?

**Modeling and understanding looping.** Next, building on Bachmann & Nagarajan (2024), we introduce a synthetic graph reasoning task with star graphs to isolate how errors in learning cause looping in a controlled setting. We train Transformers from scratch on random-walk traces that start at a designated start node and aim to reach a leaf goal, mimicking a chain of thought that sometimes makes progress and sometimes backtracks. In this setup we demonstrate two mechanisms through which learning errors cause looping.

*Risk aversion due to hardness of learning (Section 3).* We show that hardness of learning can induce a form of risk aversion. When the correct progress-making action (e.g., the next step in a proof) is hard for the model to learn, but an easy cyclic action is available (e.g., restating a previously stated fact), the model puts relatively more probability on the easy cyclic action, causing looping at low temperatures. We formalize this in Proposition 3.1, which shows that indistinguishability of the hard action diffuses its probability across many alternatives, while the easy action retains its mass. In our graph reasoning task, this leads to low-temperature loops. Temperature reduces looping and improves accuracy, but the model still assigns too little probability to the progress-making action, so chains at high temperature remain much longer than those of a perfect learner.

*Inductive bias for temporally correlated errors (Section 4).* We also show that even in the absence of hardness of learning, Transformers can have an inductive bias toward looping. When the training distribution places (nearly) equal mass on several progress-making actions at a decision point, small estimation errors tilt the model toward a few options, and these errors are correlated over time. When a similar decision point reappears later in the chain, the model tends to reselect the previously favored actions. Under greedy or low-temperature decoding, these small, temporally cor-

related errors are amplified and produce loops. We again demonstrate this mechanism in the graph reasoning task, and we observe qualitatively similar patterns in example traces from real reasoning models.

*A catalyst for looping (Section 5).* On top of these mechanisms, we discuss another force: as the model repeats the same text, its confidence in continuing the repetition grows. This acts like a catalyst for any looping mechanism.

**Role of randomness.** Across these mechanisms, while temperature reduces looping by promoting exploration, it does not fix the underlying errors in learning. As a result, generations at higher temperatures can still be longer than necessary. How effective temperature is depends on the size of these errors. When they are small, as in the temporally correlated errors setting, the increase in response length at high temperature is modest. When they are large, as in the risk-aversion mechanism, response lengths can be much larger than those of a perfect learner. We see similar behavior in real reasoning models: at higher temperatures, student models such as OpenThinker-3 on average produce longer chains than their teacher (Fig. 1). In this sense, randomness or temperature is useful but not a holistic solution.

Finally, while our synthetic reasoning task provides an effective testbed to cleanly isolate the several mechanisms, it does not capture the full complexity of real reasoning models, and more work is needed for a thorough understanding of looping in real models. We discuss the limitations of our synthetic setup in Section 6, along with several promising directions for future work, including more holistic training-time interventions that directly reduce errors in learning.

To summarize, our main contributions are: (i) pointing to learning errors as a key cause of looping via experiments on open models, (ii) demonstrating mechanisms by which learning errors cause looping, via a novel graph reasoning setup, and (iii) evidence that temperature reduces looping but is not a holistic solution.

### 1.1. Related work

While looping has been especially prevalent in *reasoning* models, it has been observed and studied since the early days of large language models (Fan et al., 2018). Holtzman et al. (2020) brought broad attention to this "neural text degeneration," showing that low-temperature sampling or beam search can yield generic and repetitive text. In response, several mitigations were explored. Unlikelihood training explicitly down-weights repeated or undesirable continuations (Welleck et al., 2020), and contrastive methods encourage more isotropic token representations, which reduces repetition (Su et al., 2022).

A key data-centric insight was that model repetitions were correlated with repetitions in the training corpus; as

instruction-tuning data improved and models scaled, looping became less severe (Li et al., 2023). This aligns with our evaluations as well: we find several instruction-tuned models exhibiting little looping.

With the rise of *reasoning* models, however, severe looping has re-emerged. The nature of chain-of-thought data, which includes cyclic actions like backtracking and reflection (Li et al., 2025; DeepSeek-AI et al., 2025; Cuadron et al., 2025; Gandhi et al., 2025) provides fertile ground for models to fall into loops. Moreover, scaling alone is not a satisfactory solution for reasoning models: a core promise of this paradigm is to leverage inference-time compute so that even small models can perform well via longer chains. Understanding and holistically mitigating looping in this setting is therefore important, and our work takes a step towards this.

**Conflict of Interest Disclosure.** C.P. was employed as an intern by Microsoft Research during the development of this work, and all other authors are full-time employees of Microsoft. Microsoft leads the development of Phi-4, which was among the open-source models evaluated in this paper.

## 2. Observations on open models

We conduct a large-scale study of looping on open reasoning models, spanning a range of model sizes and training paradigms such as distillation from a teacher and RL post-training. The models we test are as follows. **Qwen:** DeepSeek-R1 Distilled Qwen 1.5B, 7B, 32B (DeepSeek-AI et al., 2025); **OpenThinker3:** OpenThinker3 1.5B, 7B (Guha et al., 2025) and QwQ-32B (Team, 2025) (the teacher for OpenThinker-3); **Phi-4:** Phi-4-reasoning, Phi-4-reasoning-plus (Abdin et al., 2025); **Llama:** DeepSeek-R1 Distilled Llama 8B (DeepSeek-AI et al., 2025).

We mark a response as looping if it contains some $n$-gram appearing at least $k$ times. We use $n = 30$ and $k = 20$ for all reasoning models (see Appendix A.3 for ablations). All plots report averages over AIME 2024 and 2025 (see Appendix A.4 for ablations with GPQA). For each triple (problem, model, temperature) with temperature $\in$ $\{0, 0.2, 0.4, 0.6, 0.8, 1.0\}$, we sample 20 responses and compute accuracy, looping percentage, and response length, and then average across problems. We show looping percentages and response lengths as a function of temperature for the OpenThinker3 family in Figure 1. Plots for the other model families, including additional results for instruct models, are in Appendix A. Our main observations are:

**Looping decreases with temperature.** All open reasoning models we tested loop at low temperature, and looping decreases as temperature increases. This is consistent with prior work (Holtzman et al., 2020) that reported similar trends for earlier models such as GPT-2.

**Higher-capacity models loop less.** At low temperature, looping decreases with model capacity (e.g., 76% for Qwen-1.5B vs. 37% for Qwen-32B). Moreover, when distilling from a high-capacity teacher to a lower-capacity student, the student can loop much more than the teacher (e.g., 30% for OpenThinker-3 1.5B vs. 4% for QwQ-32B).

**Higher-capacity models produce shorter responses.** While looping largely disappears at higher temperatures, higher-capacity models still produce shorter responses (Fig. 1). As we argue later, this relates to temperature reducing looping without holistically fixing the underlying causes.

**Harder problems elicit more looping.** For most models, problems that are harder at high temperature tend to loop more at low temperature. In each 15-problem AIME split (24-I/II, 25-I/II), we rank problems by accuracy at temperature 0.8, take the easiest 5 and hardest 5, and pool across splits to form 20 easy and 20 hard problems. Looping at temperature 0 is then significantly higher on hard than easy problems (e.g., 51% vs. 13% for OpenThinker-3 1.5B). The only exception is Phi-4 reasoning which we discuss in Appendix A.2. This suggests that no matter how large a model is, there may always exist problems that are hard enough to induce looping, though we leave a deeper study of this conjecture to future work.

**What causes looping?** Two patterns stand out: low-temperature looping decreases with model capacity, and distilled students can loop far more than their (larger) teachers. If a student perfectly learned its teacher distribution, we would not expect it to loop substantially more than the teacher. This gap points to imperfect learning, that is, systematic errors in the learned distribution, as a key cause. The fact that most models loop more on harder problems also aligns with this view.

### 2.1. Example Traces

To qualitatively understand looping, we analyze looping traces generated by OpenThinker-3 1.5B (for detailed analysis including model outputs see Appendix B).

**Example 1.** *We prompt OpenThinker-3 1.5B to solve AIME 2025 Problem I-7 with greedy decoding. The task is: pair A–L into six two-letter words (each pair sorted alphabetically), sort the six words, and compute the probability that the last word contains G. The model begins sensibly, correctly counting the total number of pairings, but then makes little progress. It first loops semantically by repeatedly re-deriving the fact that each word is the sorted pair. Later, when it turns to pairs involving G, it collapses into a hard loop that repeats the same statement (e.g., "the word is the pair sorted") and never exits.*

*This trace illustrates a kind of risk-aversion: under greedy decoding, the model repeatedly takes easy cyclic actions*

*(e.g. repeating the same fact about how a two-letter word is formed), instead of progress-making actions, such as exploring the structural insights needed to solve the problem. In further analysis, we take the model prefix immediately before the final loop and make the teacher model (QwQ-32B) continue under greedy decoding. The teacher does not get stuck in a loop and makes substantial progress. Moreover, when sampling high-temperature continuations from the same prefix, teacher generations are far more likely to contain key progress-making insights than student generations. Overall, the student model demonstrates risk-aversion under greedy decoding, preferring easy cyclic actions even when the teacher can make progress. We provide a detailed quantitative analysis of this prefix, including probability mass on progress-making actions for both student and teacher, in Appendix B.1.*

**Example 2.** *We observe a different looping pattern on AIME 2025 Problem I-5. Here the task is to count the $8!$ permutations of digits $\{1, \ldots, 8\}$ that form an eight-digit integer divisible by $22$. The model correctly reduces divisibility by $22$ into divisibility by $2$ and $11$ and notes the last digit must be even. However, when applying the $11$-divisibility condition, it makes a mistake that eventually yields a contradiction. At that point the model restarts the analysis, but makes the same mistake, reaches the same contradiction again, and repeats this series of steps several times. Unlike risk-aversion loops, where the model is averse to making forward progress, here it does move forward but repeatedly makes the same mistake, leading to a loop.*

The analysis in the last subsection pointed to errors in learning as a key cause of looping. In the above examples, we saw qualitative looping patterns such as risk aversion and repeatedly making the same mistake. How do errors in learning actually lead to these patterns? Understanding this directly is challenging: it would require characterizing the teacher's data distribution, which can be very complex for real reasoning models, and then analyzing how training changes this distribution in the student, which adds further complexity. To make progress, we therefore consider a synthetic graph reasoning task and use it to demonstrate looping mechanisms in a controlled setting. In particular, Section 3 shows how errors in learning can produce risk-aversion similar to Example 1, while Section 4 shows looping due to temporally correlated errors similar to Example 2.

## 3. Risk-aversion due to Hardness of learning

We first explain how hardness of learning can lead to risk aversion and looping, and then describe the graph reasoning task, which we use to instantiate this looping mechanism. It is useful to keep in mind the distillation scenario, where a student model is trained on traces generated by a teacher (we later discuss how these ideas extend beyond distillation).

As an example, suppose that at some step in the reasoning chain, the teacher's data distribution has support over two actions: a progress-making action (e.g., the next logical proof step) and a cyclic action (e.g., backtracking or repeating a fact). Assume the progress-making action is hard for the student to learn, while the cyclic action is easy. We show that even if the teacher distribution assigns high probability to the progress-making action, the student can still place relatively more mass on the cyclic action. Thus, while greedy decoding under the teacher distribution would lead to progress, greedy decoding with the student tends to pick the cyclic action repeatedly and get stuck.

The mechanism has two components: (i) a hard-to-learn action co-occurring with an easy action, which increases the easy action's relative probability and leads greedy decoding to prefer it, and (ii) the easy action being cyclic, which, together with its relatively high probability, causes the model to repeatedly select it. We formalize the hardness of an action below and defer the formalization of cyclic actions to the next subsection, after we introduce the graph task.

**Formalizing hardness.** We say an action is hard if the model cannot distinguish it from $n$ other actions (e.g., the correct next proof step is confusable with $n$ alternatives). A larger $n$ corresponds to a harder action. In this case, even if the training distribution assigns high probability to the hard action, the model, trained to maximize log-likelihood, diffuses that mass across the indistinguishable options. The easy action then ends up with relatively higher mass. One way to formalize this is as follows:

**Proposition 3.1.** *Consider the following task: there exist $n$ sets of contexts $C_1, \ldots, C_n$ which are equi-likely under the training distribution. And there are $n$ distinct "hard" actions $a_1, \ldots, a_n$, and an "easy" action $a_0$. For every context $c_i \in C_i$, the training distribution picks action $a_i$ with probability $(1 - p)$ and $a_0$ with probability $p$. Now consider a learner that cannot distinguish between the $n$ hard actions or, in other words, it is constrained to ignore the context when deciding on the best action. Then the maximum log-likelihood solution for such a learner assigns probability $p$ to the easy action $a_0$ and probability $(1-p)/n$ to the hard indistinguishable actions $a_i, \forall i \in \{1, \ldots, n\}$.*

We provide the proof in Appendix C, and also discuss there how the proposition applies to language models. To appreciate the implications, note that as the action becomes harder (i.e., as $n$ increases), the probability assigned to it decreases as $(1 - p)/n$, while the probability of the easy action remains $p$. Thus, for sufficiently large $n$, greedy decoding picks the easy action.

More generally, whenever the student cannot reliably distinguish an action from several plausible alternatives, maximum-likelihood training encourages it to hedge by spreading probability mass across these candidates. Propo-

sition 3.1 captures this effect in an extreme case where the hard actions are completely indistinguishable, but the same qualitative behavior can arise whenever the student's representation collapses many distinct contexts, forcing the model to diffuse probability mass across the actions that are plausible in those contexts. This collapse can arise for several reasons, including limited capacity of the student model or optimization difficulties during training.

### 3.1. Demonstration with graph reasoning

We demonstrate the looping mechanism discussed above in a synthetic graph reasoning task, which is a convenient testbed for studying looping: it lets us precisely define notions such as progress-making and cyclic actions, as we elaborate later in this section.

**The star graph.** Our graph reasoning task builds on the hardness result of Bachmann & Nagarajan (2024), who train Transformers to find paths in a star graph. A star graph $G(n, \ell)$ is a directed graph with a *root* $r$ and $n$ simple "spokes," each a path of length $\ell - 1$ ending at a distinct leaf (Figure 2). Each training ex-

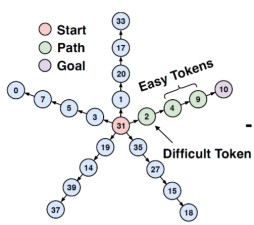

*Figure 2.* Illustration of star graph. Figure from Bachmann & Nagarajan (2024).

ample is a sequence containing the edge list, a start node (the root $r$), a goal node $g$ (a leaf chosen uniformly at random), and a path from $r$ to $g$. Distinct instances are formed by randomly permuting node labels. Bachmann & Nagarajan (2024) show that Transformers trained with next-token prediction fail to learn the correct path.

Here, aside from the root, all nodes have a single outgoing edge. Thus, for path-finding, learning the first edge on the path is harder than learning the remaining edges. Bachmann & Nagarajan (2024) show that models learn the later edges early in training; once those are learned, the first edge becomes the bottleneck. As a result, the learned solution places roughly $1/n$ probability on each outgoing edge from the root while predicting subsequent edges correctly, yielding chance accuracy (see Hu et al. (2025) for a proof of hardness). Note that here the source of hardness is not model capacity (the models have enough capacity to represent the right solution), but the inability of optimization.

**The graph reasoning task.** Our graph reasoning task is obtained by making two modifications to the above setup.

*First*: we introduce an explicit start node $s$ with a single outgoing edge to the root $r$. The rest of the graph remains as in the standard star graph, and the goal $g$ is still a leaf. This modification is not crucial for our results, but it helps illustrate the mechanism better (as discussed below).

*Second (more crucial)*: we train on a random-walk trace rather than a single path. The walk starts at $s$. At any node, it moves one step toward $g$ with probability $1 - p$ and resets to $s$ with probability $p$. For simplicity, this reset applies at every node, including $s$. When $p = 0$, this reduces to the original single-path setting (with the added start node). We use $p = 0.3$.

We abuse notation and use $G(n, \ell)$ to denote this modified star-graph where $n$ paths of length $\ell - 1$ emanate from the root, and there exists a separate start node with a single outgoing edge to the root. Finally, in some experiments, we also add a small exploration probability, described later. An example training instance is shown below:

$$\underbrace{\texttt{5,42|2,29|...|29,22}}_{\text{edge list}} \; / \; \underbrace{\texttt{2,42}}_{\text{start, goal}} = \underbrace{\texttt{2,29,22,...,5,42}}_{\text{random walk}}$$

Each instance is a sequence containing an edge list, a start node, a goal node, followed by a random walk from start to goal. Each node is a separate token, and $\{\,|\,, /, =\}$ are separator tokens. As in the original star-graph setting, distinct instances are formed by randomly permuting node labels.

The notions of actions, cyclic actions, and progress-making actions can be precisely stated in this graph reasoning task. An *action* is simply the choice of the next node to visit from the current node. A *cyclic action* is an action that takes the walk to an already visited node, and a *progress-making action* is an action that moves the walk closer to the goal node. In the above task, the teacher distribution takes the progress-making action of moving forward to the next node toward the goal with probability $0.7$ and takes a cyclic action (reset to the start) with probability $0.3$.

For a language model, an action can be viewed as a short chunk of tokens implementing a logical step. Thus if one views the source-to-goal path in the original star-graph setting as a model for the chain of thought of earlier language models that moves toward the goal in a step-by-step manner, then the random-walk variant can be seen as a model for reasoning language models that explore multiple strategies, backtrack, and restart. Our aim, however, is not to capture the full complexity of an actual reasoning model, but to isolate the phenomenon in the simplest setting possible.

**Intuition via a Markovian student.** The random walk defined above only visits nodes along the path from the start to the goal. At each visited node, the training distribution places probability $0.7$ on the progress-making action and $0.3$ on the cyclic action (reset to the start). A perfect student that exactly recovers this distribution would, under greedy decoding, always take the progress-making action and reach the goal without looping. However, hardness at the root breaks this behavior and induces looping.

The reset action is easy to learn: it only requires reading the start node from context and jumping to it. The progress-

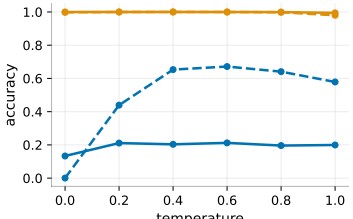 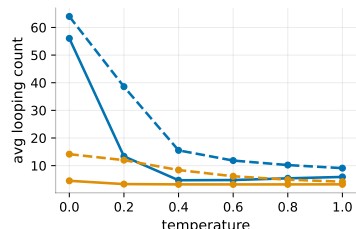 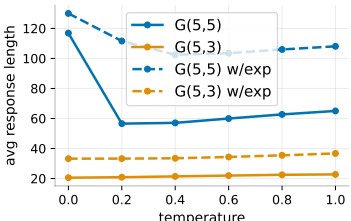

*Figure 3.* **Hard-to-learn actions induce low-temperature loops.** We train small Transformers on random-walk traces on star graphs $G(5,5)$ (hard) and $G(5,3)$ (easier). In the training distribution, the progress-making action is taken with probability 0.7 and the reset action with 0.3; dashed curves use an exploration variant at the root (0.5 correct edge, 0.2 other children, 0.3 reset). The looping count is the average number of root → start transitions per trace; a perfect learner would have looping count 0 at $T$=0. On $G(5,5)$, looping and response length are large at low temperature and much smaller at higher temperatures; without exploration, accuracy stays near chance, while with exploration it is near 0 at $T$=0 and substantially higher at higher temperatures. On $G(5,3)$, looping and response length are small and almost flat, and accuracy is near-perfect for all temperatures. Overall, a hard progress decision at the root, paired with an easy cyclic reset action, drives low-temperature loops, and making the problem easier or increasing temperature reduces them.

making action is also easy at all nodes except the root, where the student must choose among $n$ outgoing edges. If it cannot distinguish these $n$ options, we expect it to spread the 0.7 mass on the progress-making action uniformly, assigning $0.7/n$ to each outgoing edge, while still assigning 0.3 to the easy reset action.

Now suppose that the student is Markovian, so that its next-action distribution depends only on the current node and not on the full history of the walk. Under greedy decoding, the distributions above imply that the walk will loop between the start node and the root: from the start, the student moves to the root, and from the root, whenever $0.3 > 0.7/n$, it resets back to the start. Note that the student does not get trapped at the start node: both "move to the root" and "stay/reset" are easy to learn there and are assigned probabilities close to 0.7 and 0.3. Loops arise when a hard action at the root coexists with an easy cyclic reset, and we introduce the explicit start node to make this contrast clear.

### 3.2. Observations with Transformers

The discussion above assumes a Markovian student, but Transformers can condition on the full prefix. We therefore train a decoder-only Transformer with $\approx 85M$ parameters from scratch and ask whether similar looping emerges (see Appendix D for training details).

**Low-temperature looping.** On $G(5,5)$, we do observe looping similar to the Markovian student discussed above. Figure 3 reports accuracy, response length, and the *looping count* (the number of root → start transitions per test instance) as a function of temperature. Both looping count and response length are large at low temperature and much smaller at higher temperatures; at $T$=0, the looping count is roughly half the response length since traces often alternate start → root → start → root . . .. Eventually the model commits to an outgoing edge from the root and walks deterministically to a leaf, but because it cannot reliably identify

the correct edge, accuracy remains near chance.

**Accuracy improves with temperature.** While both looping count and response length are much smaller at higher temperatures, accuracy stays near chance across temperatures. This contrasts with many reasoning models, where accuracy often improves as temperature increases. The difference stems from the training distribution: the walk never explores off-path routes, so the model learns to revisit the *same* path after each reset. Since it cannot reliably pick the correct path at the root, accuracy remains at chance even when loops shorten.

To test exploration, we modify the walk at the root: on each root visit, the walk resets with probability 0.3, takes the correct outgoing edge with probability 0.5, and takes one of the other edges uniformly with probability 0.2; all other internal transitions remain the same (and reaching a non-goal leaf triggers a reset with probability 1). Training on these traces makes accuracy substantially higher at higher temperatures than at $T$=0 on $G(5,5)$ (w/exp in Figure 3). As in the non-exploration setting, looping count and response length are much smaller at higher temperatures. As a side note, at $T$=0 the model often bounces between start and root and eventually terminates rather than committing to a path (so accuracy can be near 0). At very high temperatures, accuracy degrades slightly because the model occasionally samples tokens corresponding to invalid edges in the graph, and more walks run out of the token budget without reaching the goal. Also, response lengths are somewhat higher than in the non-exploration setting overall, likely because exploration produces longer training traces and the model mirrors this at test time.

**Less looping on easier problems.** We also evaluate $G(5,3)$ (with and without exploration). Since the paths from the root to the leaves are shorter, the progress-making action at the root is less hard to learn. Indeed, the learned probabilities place higher mass on the correct outgoing edge from the

root than on the others. However, early in generation the model often still prefers the reset at the root over the correct outgoing edge (especially in the exploration variant), so brief looping occurs before it commits to the correct edge and reaches the goal. As a result, accuracy is near perfect across temperatures, and the average response length is essentially stable. This mirrors our earlier finding from Section 2: models loop less on easier problems.

**Overestimation of the reset action.** We expected that at the root the model roughly preserves the teacher's 0.3 reset probability while spreading the remaining 0.7 uniformly over outgoing edges. In practice, the trained Transformer often overestimates reset. For example, on $G(5,5)$ (with exploration) at temperature 0, across 4000 generated trajectories, at $\approx 91\%$ of root positions the model assigns reset probability above 0.3 (mean $\approx 0.33$), while spreading the remaining mass roughly uniformly over the outgoing edges. This further strengthens the cyclic action and exacerbates looping. We see a similar bias on $G(5,3)$, which explains brief initial looping there even though the model puts more mass on the correct root edge than on the others. We hypothesize that this overestimation bias arises from the prevalence and ease of learning of the reset action (neural nets are known to prefer easier functions (Kalimeris et al., 2019; Rahaman et al., 2019)), but leave a thorough analysis to future work. Overall, two forces appear to contribute to the dominance of the easy reset action over the hard progress action: the spreading of progress-making mass over the $n$ outgoing edges, and the over-estimation of the reset action.

### 3.3. Other implications

**Sources of hardness.** In the illustration above, hardness comes from the inability of the optimization to find the right solution. More generally, hardness can arise due to other factors too such as limited model capacity or limited training compute (under-training). The hardness due to model capacity is a plausible explanation for why smaller models loop more within a family trained on the same data. In Appendix G, we demonstrate hardness due to model capacity causing risk-aversion looping in our graph reasoning setup.

**Temperature as a mitigation.** We began by asking whether temperature is a stopgap or a holistic fix. The experiments above suggest an answer. Increasing temperature reduces looping, but it does not remove the underlying learning errors: the model still assigns too little probability to the correct progress-making action. One simple way to see this is response length. On $G(5,5)$ with exploration, at temperature 1 the learned model has average response length 108.1, versus 24.4 for a perfect learner (a $\approx 4.4\times$ blow-up). Even when looping is reduced, the model still takes many extra steps because it has not learned the right action probabilities. We saw a similar trend with open reasoning models, where smaller models produced longer chains than

larger models or their teachers. The same issue also shows up in accuracy: with a similar token budget, the learned model reaches only about $58\%$ at temperature 1, while the teacher achieves near-perfect accuracy with the same token budget. Thus, temperature can reduce looping at the surface level, but the underlying cause remains. We discuss more holistic training-time interventions in Section 6.

Finally, Appendix D discusses further implications of these experiments, including comparisons between reasoning vs. instruct models and distilled vs. RL-trained models.

## 4. A Bias for Temporally Correlated Errors

In this section, we show that Transformers have an inductive bias for looping, causing its errors to correlate over time, similar to Example 2. The mechanism is easiest to describe in the graph reasoning setting, so we directly jump in.

**Setup.** We use the same star graph and training setup as in Section 3, but change the teacher walk transitions to remove hardness at the root and remove per-step resets. The walk starts at the start node, moves to the root, chooses a root child uniformly at random, and then follows the unique outgoing edges down that spoke to a leaf. If the leaf is the goal, the walk stops; otherwise it resets to the start and repeats. Training and evaluation match the previous section.

### 4.1. Observations and implications

**Low temperature loops.** It helps to first consider a perfect learner. On $G(n,\ell)$, such a model would learn: probability 1 to the unique outgoing edge at non-root internal nodes; probability 1 to reset at non-goal leaves; and probability $1/n$ to each child at the root. In practice, the trained model deviates slightly at the root. Instead of learning an exact $1/n$ split, it makes small errors (e.g., $0.2 \pm 0.05$ for $n{=}5$). More importantly, these errors are *correlated across time*: the root children that are slightly preferred early in the trace tend to remain preferred on later visits to the root. Under greedy decoding or low temperature, the model therefore keeps revisiting the same one or two paths, producing loops. This is qualitatively similar to Example 2, where the model repeatedly restarts after a contradiction but re-enters the same mistaken reasoning path.

We quantify this with a *looping count*: for each generated trace, we record how many times each root child is visited and report the maximum over children; we then average this value over 4000 test instances. We observe that the looping count is high at low temperature and decreases as temperature increases (Fig. 4, margin $= 0$). The average response length shows the same trend. Accuracy is near chance at temperature 0 and improves with temperature.

**Temporal correlation.** In addition, we directly probe temporal correlation in the model's predictions at the root. We

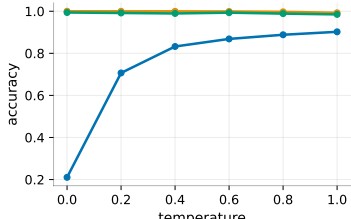 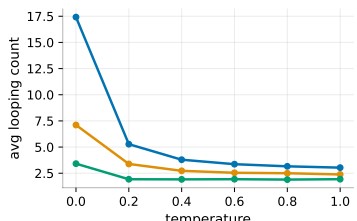 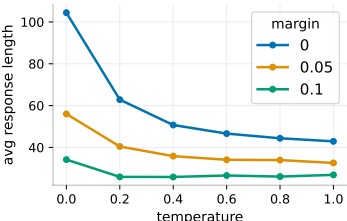

*Figure 4.* **Temporally correlated errors induce low-temperature loops.** We train on $G(5,5)$ star-graph random-walk traces that choose a child at the root, take the unique outgoing edge at other internal nodes, and reset to the start at non-goal leaves. For $m{=}0$, the training walk always chooses a root child uniformly at random; for $m{>}0$, a margin biases the walk toward children that have not yet been visited. The trained models develop *temporally correlated* errors at the root, repeatedly selecting the same few children at low temperatures and thus looping; this effect is weakened, but not removed, by the margin. Accuracy: near chance for $m{=}0$ at low temperature and increasing with temperature; margin variants achieve near-perfect accuracy across temperatures. Looping count: for each trace, we count visits to each root child and take the maximum, then average over test instances. Lower temperatures have higher looping counts; margins reduce them. Response length: longer at lower temperatures, and shortened by margins.

measure two quantities. For each generated trace, we look at every pair of *consecutive* visits to the root and, for each pair, we find the two children with the highest probabilities. We then compute (i) the fraction of such consecutive-visit pairs where the child with the highest probability is the same at both visits, and (ii) the analogous fraction for the child with the second-highest probability. These fractions are $98.7\%$ and $72.8\%$, respectively—far higher than expected if predictions on consecutive visits were uncorrelated. Thus the Transformer's errors at the root are temporally correlated, inducing an inductive bias for looping.

**A variant with margins.** The training walk above samples root children uniformly at every visit. What if the training distribution itself discourages revisiting already-seen children? We study a *margin* variant: the first time the root is visited, a child is sampled uniformly; on later visits, each child that has ever been visited at the root has its sampling probability reduced by a fixed margin $m$, and the removed mass is redistributed uniformly over the as-yet-unvisited children. This reduction is applied once per child (its probability does not decrease further with repeated visits). With $m{=}0$ we recover the random walk considered above.

Training with margins ($m{=}0.05$ and $m{=}0.1$ ) reduces looping, but it remains non-trivial (Fig. 4). Temporal correlation also weakens but remains substantial: the consecutive-visit agreement for the top (and second) child drops from $98.7\%$ ($72.8\%$) at $m{=}0$ to $75.3\%$ ($52.7\%$) at $m{=}0.05$ and $52.5\%$ ($40.1\%$) at $m{=}0.1$. Importantly, under the margin-modified training distribution, a perfect student decoded greedily would never revisit an already-visited child at the root, so the corresponding agreement fraction for the top child would be $0\%$. Thus, non-trivial looping driven by temporally correlated errors persists even when the teacher distribution explicitly discourages revisiting past actions.

**Comparing the mechanisms.** Both mechanisms we discuss stem from errors in learning, but they differ in na-

ture. The first arises from *hardness of learning*: probability mass on a hard progress-making action is spread across many indistinguishable alternatives. This gap can be large relative to the training distribution and does not rely on Transformer-specific inductive biases—it can appear for any maximum-likelihood learner when the correct action is indistinguishable from many others. The second relies on an *inductive bias toward temporally correlated errors*: at repeated decision points, the learned probabilities are slightly skewed toward a few actions. Here the deviations are small, yet sufficient for low-temperature looping.

Temperature helps in both settings mainly by promoting exploration, but it does not fix the underlying learning errors, so generations at high temperature can still be longer than those of a perfect learner. In the hardness setting (Section 3), this length blow-up is large ($\approx 4.4\times$). In the present setting, the errors are smaller: the model's distribution at the root is already close to the ideal with only mild skew. On $G(5,5)$ with margin $=0$ at temperature 1, the learned model has response length $42.9$ compared to $29.6$ for a perfect learner (a $\approx 1.5\times$ blow-up). This suggests that temperature is a more effective fix when learning errors are small, while larger errors may require more direct training-time interventions.

These mechanisms can also be viewed as two atomic "forces" driving looping. In Appendix E, we present a variant where they act together.

## 5. A Catalyst for Looping

In real model traces, we also see another interesting phenomenon. At the beginning of a loop, the model's probability distribution over next tokens looks relatively normal, but as it repeats the same text, it becomes increasingly confident (Figure 16). This acts as a catalyst for any looping mechanism, making it harder for the model to escape once it has been looping for a while. This behavior has also been

observed in prior work for earlier generations of language models (Holtzman et al., 2020; Chiang & Chen, 2021; Xu et al., 2022), and can also be viewed as an instance of error accumulation or self-conditioning (Sinha et al., 2026).

Using our synthetic graph reasoning setup, we show that a tiny fraction of looping traces during training can produce such a buildup. Specifically, in the risk-aversion setup (Sec. 3), we modify the training distribution: with probability 0.001 the example is a trace that deterministically loops between the start and root, otherwise sampled as before. This probability is small enough not to noticeably change the learned distribution early in a generation, but once the model has already looped, conditioning on this history makes continuing the loop much more likely (Fig. 5). Analogous results for the other mechanism are in App. F.

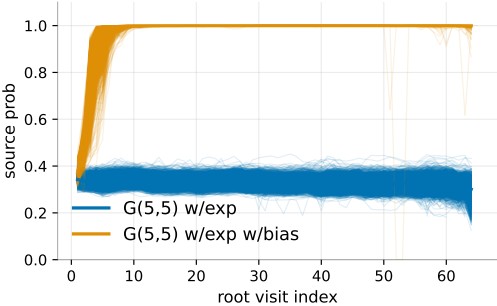

*Figure 5.* **Tiny bias drives reset probability toward 1.** In the risk-aversion setting on $G(5,5)$ with exploration, we record at every root visit the model's probability of taking the reset-to-start action, across 4000 greedy-decoded traces (thin lines; thick line is the average). When training includes a tiny fraction (0.001) of deterministic looping traces (orange), the reset probability rises toward 1 as the loop persists, making escape increasingly unlikely. Without this bias (blue), the probability stays roughly constant.

In reasoning models, the underlying bias can arise from various sources. For example, during pretraining the model may see a few sequences where some text is repeated many times. As a result, during inference, once the model has already repeated itself, the conditional probability of continuing the repetition can become larger than its unconditional value.

## 6. Discussion

We began with two questions: why do reasoning models loop, and is temperature a holistic fix or a stopgap? Our evaluation of open reasoning models points to errors in learning as a key cause. Using a synthetic graph reasoning task, we identified two mechanisms by which learning errors cause looping: risk aversion caused by hardness of learning, and an inductive bias toward temporally correlated errors.

On the role of randomness, temperature is reasonably effective at reducing looping, but does not fix the underlying learning errors: the model still assigns too little probability

to progress-making actions, so response lengths at high temperature remain longer than necessary. Thus temperature is a reasonable fix when errors are small (temporally correlated errors mechanism), but a stopgap when they are large (hardness-based mechanism).

### 6.1. Limitations and Future Work

**Synthetic-to-real mapping.** Our synthetic graph reasoning task is intentionally simple, designed to isolate mechanisms behind looping. The examples discussed in Section 2.1 show glimpses of these mechanisms in real reasoning model traces. For example, in Example 1 we observe that at the looping point, the teacher model places substantially more mass on the progress-making action than the student, consistent with the risk-aversion mechanism (see Appendix B.1 for the full quantitative analysis). While these examples are consistent with the proposed mechanisms, they admittedly do not capture all the details. For instance, beyond showing the teacher puts more mass on progress-making actions, we do not precisely characterize all possible actions and the mass on each. Such precision is challenging for two reasons. First, actions in real models live in semantic space, making it hard to cluster continuations into discrete actions with reliable probability estimates. Second, multiple forces act simultaneously and are hard to disentangle. For example, within risk-aversion, our synthetic experiments reveal two sub-forces (probability diffusion across indistinguishable alternatives, and overestimation of the easy cyclic action), both consistent with the progress-making action receiving low mass but having different preferences for distributing the overall mass (see Appendix E and F for other examples of forces interacting). These challenges motivate our use of synthetic tasks; developing a tighter mapping between real models and our setup is an important future direction.

**Other forces.** Further, given the complexity of real models, other forces likely also contribute to looping. One example is *error accumulation*: errors made during generation can compound over time. Our proposed mechanisms show that looping can arise even without such accumulation, but it can act as a catalyst—the self-conditioning effect of Section 5 is one instance. We also observe more general variants in real model traces (Appendix B). Pinning down such forces and their interactions is an important future direction.

**Beyond temperature.** For mitigation, a natural next step is to seek holistic fixes that directly reduce learning errors. Some errors are inevitable due to capacity, but it is unlikely that all of today's errors are necessary. One concrete direction is to modify how we distill teacher traces, making them easier for the student to learn. For example, a targeted data augmentation approach could identify points in teacher traces that the student finds hard (e.g., high-loss positions) and augment them with brief hints.

## Acknowledgements

We would like to thank the anonymous reviewers for their constructive feedback that helped improve our paper. Additionally, we are deeply grateful to Vaishnavh Nagarajan for his sharp and detailed comments on our work.

## Impact Statement

This paper presents work whose goal is to advance the field of Machine Learning. There are many potential societal consequences of our work, none which we feel must be specifically highlighted here.

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

# A. Additional experiments and details for Section 2

**Evaluation details.** The reasoning models we test are as follows. **Qwen:** DeepSeek-R1 Distilled Qwen 1.5B, 7B, 32B (DeepSeek-AI et al., 2025); **OpenThinker3:** OpenThinker3 1.5B, 7B (Guha et al., 2025) and QwQ-32B (Team, 2025) (the teacher for OpenThinker-3); **Phi-4:** Phi-4-reasoning, Phi-4-reasoning-plus (Abdin et al., 2025); **Llama:** DeepSeek-R1 Distilled Llama 8B (DeepSeek-AI et al., 2025). While we mostly focus on reasoning models, we also test a few instruction tuned variants (non-reasoning models). These include Qwen2.5-Math-1.5B-Instruct (Yang et al., 2024b), Qwen2.5-1.5B-Instruct (Yang et al., 2024a), Phi-4 (Abdin et al., 2024), Llama-3.1 8B Instruct (Grattafiori et al., 2024).

We consider a text response to contain looping if it contains any $n$-gram at least $k$ times. We choose $n = 30$ and $k = 20$ for all reasoning models. Since instruct models produce shorter responses, we relax the looping definition to have $k = 10$ for them.

We used the Eureka ML Insights Framework (Balachandran et al., 2024; 2025) to conduct the evaluations on all models. For the inference parameters, we kept the default `top_k` and `top_p` for each model, taken from its HuggingFace repository. We do not use any repetition penalty as we focus on the role of temperature. We ran all reasoning models with a 30K `max_tokens` budget and all instruct models, we set it based on the max allowed number of tokens for each model (14k for Phi-4, 3k for Qwen2.5-Math-1.5B-Instruct, 30k for Llama-3.1-8B-Instruct and Qwen2.5-1.5B-Instruct).

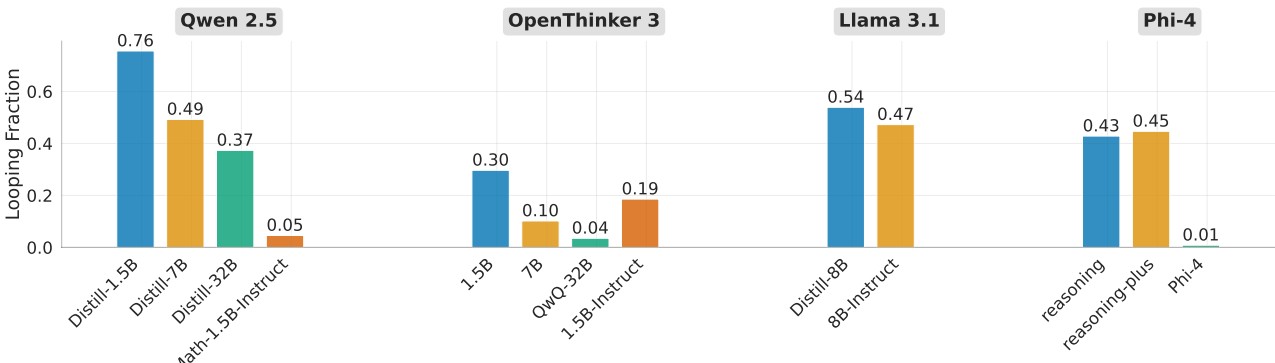

*Figure 6.* **Looping with greedy decoding.** Bars show the looping fractions at temperature 0, averaged over AIME 2024 and 2025. All reasoning models exhibit looping, and within each family larger models loop less (e.g., Qwen 1.5B > 7B > 32B; OpenThinker3 1.5B > 7B > QwQ-32B). Distilled students can loop significantly more than their teacher (OpenThinker3 vs. QwQ-32B). Reasoning models can also loop heavily even when their instruction-tuned counterparts barely loop (e.g., Qwen2.5 and Phi-4 families). Finally, RL post-training has limited effect on looping in the Phi-4 family (Reasoning vs. Reasoning-Plus).

## A.1. Other Observations

**Reasoning models can loop even when their instruct counterparts barely loop.** For the Qwen and Phi-4 model families, we see that the base instruction-tuned models loop very little, while their reasoning counterparts loop a lot. It is not always the case that the instruct models do not loop, as we see with Llama-3.1-8B-Instruct and Qwen2.5-1.5B-Instruct, but even there we observe an increase in the looping percentage for the corresponding reasoning models. The amount a model loops is likely a function of cyclic actions like restatement or backtracking in the training data. This view is supported by past work (Li et al., 2023). We discuss this further in Appendix D.

**Effect of RL training.** Phi-4 reasoning is one model family where we have access to both a distilled model (Phi-4-Reasoning) and a version that has been lightly tuned with RL (Phi-4-Reasoning-Plus). In this setting, we observe that the looping counts remain roughly the same across the two models.

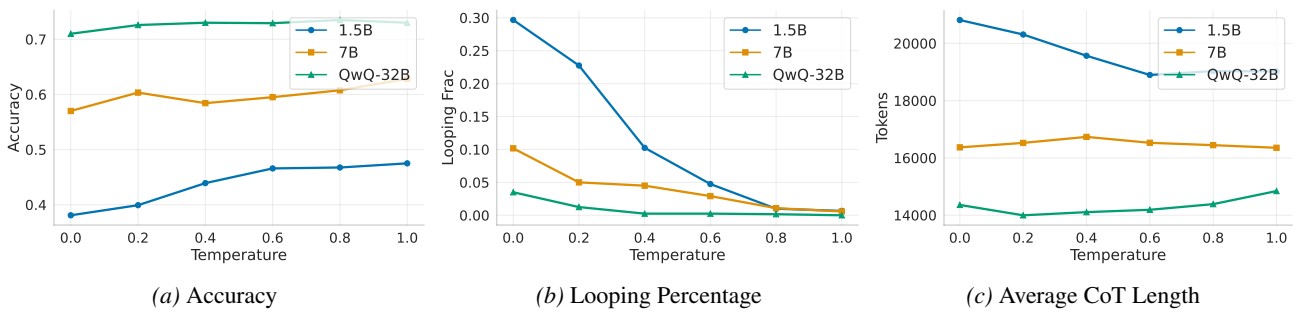

*(a)* Accuracy      *(b)* Looping Percentage      *(c)* Average CoT Length

*Figure 7.* Openthinker metrics as a function of temperature.

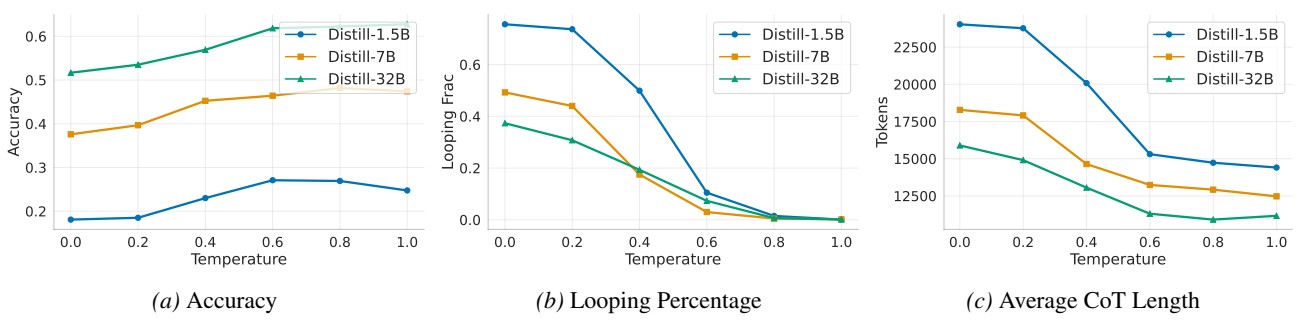

*(a)* Accuracy      *(b)* Looping Percentage      *(c)* Average CoT Length

*Figure 8.* Qwen metrics as a function of temperature.

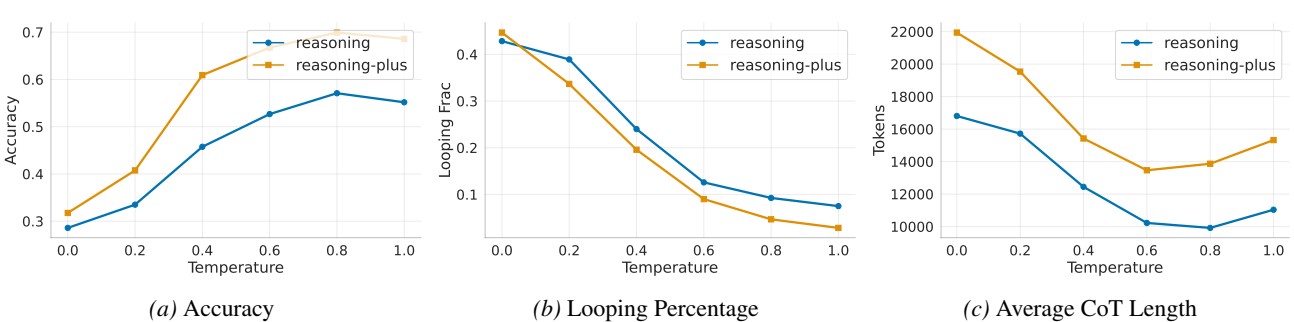

*(a)* Accuracy      *(b)* Looping Percentage      *(c)* Average CoT Length

*Figure 9.* Phi-4 metrics as a function of temperature

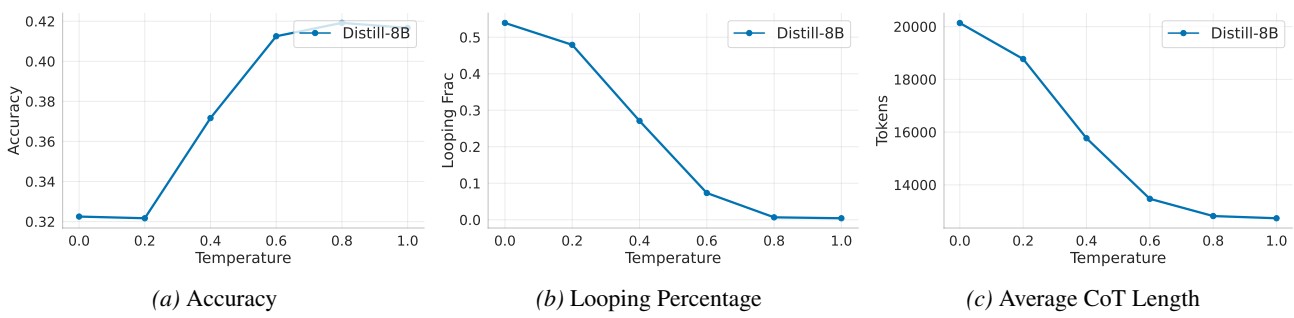

*(a)* Accuracy      *(b)* Looping Percentage      *(c)* Average CoT Length

*Figure 10.* Llama metrics as a function of temperature.

## A.2. Hard vs Eay Problems

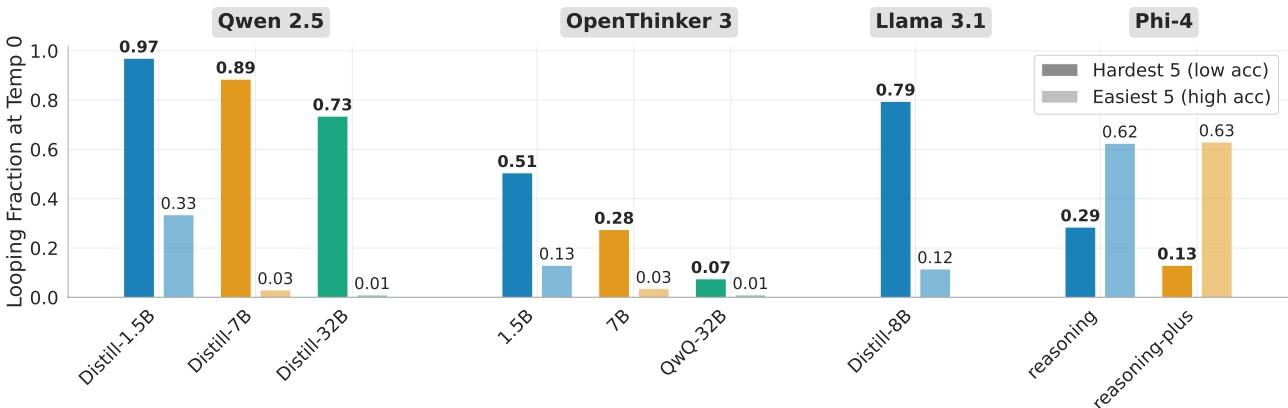

*Figure 11.* **Looping with greedy decoding for a hard-easy split of AIME problems.** For each model, we use *accuracies at temperature 0.8* to create a split of the 5 easiest problems and 5 hardest problems. Then we evaluate the *looping percentage at temperature 0* separately on these two sets of problems. The bars on the left show the looping percentage on hard problems and bars on the right show the percentage on easy problems. We can see that for all model families except Phi-4, models looping significantly more in hard problems than in easy ones.

**Phi-4 reasoning models on hard and easy problems.** The Phi-4 reasoning family of models was the only exception to the observation that hard problems induce more looping than easy ones. As we can see in Figure 11, both phi-4-reasoning and phi-4-reasoning-plus consistently exhibit more looping in easier problems than in hard ones. By manually inspecting its responses, we realized that it very often demonstrates a peculiar form of looping; it finds the answer during the CoT, then proceeds to present it, and then it gets stuck indefinitely repeating things like "We'll produce answer in plain text." This is one of the main ways in which it loops, which means that for easy problems, it will reach the solution more frequently and, thus, get stuck in this situation more often than in harder problems. Note also that another key way in which phi-4-reasoning models differ from the other models we tested is that it has been fine-tuned on OpenAI o4-mini data (Abdin et al., 2025), as opposed to DeepSeek-R1 or Qwen. Nevertheless, this is only a preliminary attempt at explaining this discrepancy and a more thorough study of the exact underlying factors would make a great direction for future research.

## A.3. Ablations on the $(n, k)$ definition of looping

In this section we ablate the $(n, k)$ definition of looping, each time keeping either $n$ or $k$ fixed and varying the other parameter. The results for different parameter settings are shown in Table 2. Note that looping percentages do not have any significant differences and the relative ordering of all models remains the same in all cases. This is not very surprising, considering our original definition of $n = 30, k = 20$ is already very strict; to consider a text as containing looping it must contain a 30-gram at least 20 times.

| Model | (20,20) | (30,20) | (30,30) | (30,60) | (40,20) |
|---|---|---|---|---|---|
| R1-Distill-Qwen-1.5B | 0.76 | 0.76 | 0.75 | 0.72 | 0.75 |
| R1-Distill-Qwen-7B | 0.51 | 0.49 | 0.48 | 0.41 | 0.49 |
| R1-Distill-Qwen-32B | 0.40 | 0.37 | 0.33 | 0.22 | 0.36 |
| OpenThinker3-1.5B | 0.33 | 0.30 | 0.28 | 0.27 | 0.29 |
| OpenThinker3-7B | 0.12 | 0.10 | 0.09 | 0.09 | 0.09 |
| QwQ-32B | 0.04 | 0.04 | 0.03 | 0.03 | 0.03 |
| R1-Distill-Llama-8B | 0.56 | 0.54 | 0.51 | 0.41 | 0.53 |
| Phi-4-reasoning | 0.46 | 0.43 | 0.42 | 0.41 | 0.42 |
| Phi-4-reasoning-plus | 0.49 | 0.45 | 0.44 | 0.42 | 0.44 |

*Table 1.* Looping percentages at temperature 0 for different (n, k) combinations

To further test whether the $(n, k)$ definition needs to be that strict, we computed looping percentages at temperature 0 for the 10-gram setting in the Qwen and OpenThinker families, with different thresholds $k$.

| Model | (10,10) | (10,20) | (10,40) | (30,20) (orig.) |
|---|---|---|---|---|
| R1-Distill-Qwen-1.5B | 0.89 | 0.80 | 0.77 | 0.76 |
| R1-Distill-Qwen-7B | 0.74 | 0.58 | 0.52 | 0.49 |
| R1-Distill-Qwen-32B | 0.64 | 0.47 | 0.41 | 0.37 |
| OpenThinker3-1.5B | 0.84 | 0.59 | 0.38 | 0.30 |
| OpenThinker3-7B | 0.72 | 0.37 | 0.16 | 0.10 |
| QwQ-32B | 0.49 | 0.16 | 0.06 | 0.04 |

*Table 2.* Looping percentages at temperature 0 for different (n, k) combinations

Even though looping numbers remain very close even with $n = 10, k = 40$, we obseve that with $n = 10$ and $k = 10$ or $k = 20$ the absolute looping increases significantly. Manual inspection of responses flagged by $(10, 10)$ but not $(30, 20)$ showed nearly all were false positives (e.g., benign reuse of expression $(33/(2\sqrt{14}))y$), motivating our use of a somewhat strict definition for looping.

### A.4. Looping percentages on GPQA

To investigate whether our observations from the AIME dataset transfer to different domains, we ran an ablation with Openthinker3 and Qwen2.5 models on the GPQA Diamond dataset (Rein et al., 2024), which contains a set of 198 highly challenging questions from biology, physics, and chemistry. For each (model, problem, temperature) tuple we generate 4 responses , for a total of $\approx 800$ responses at each (model, temperature). We evaluate on temperatures 0.0, 0.4, 0.8. We see that our main observations also hold here: lower-capacity models loop more and student models loop significantly more than teacher.

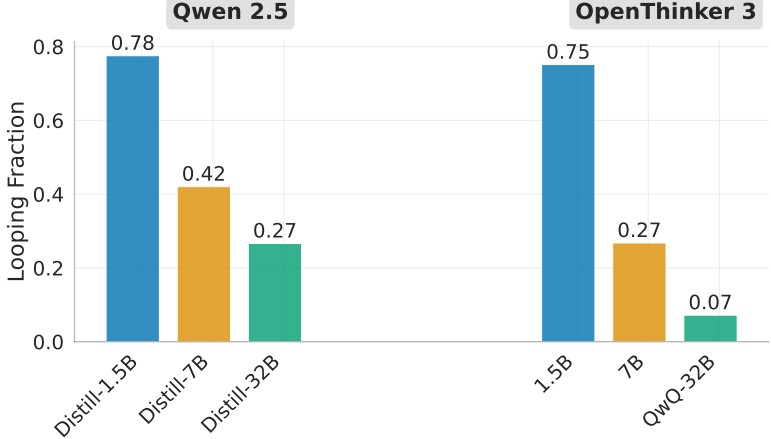

*Figure 12.* Looping with greedy decoding on GPQA.

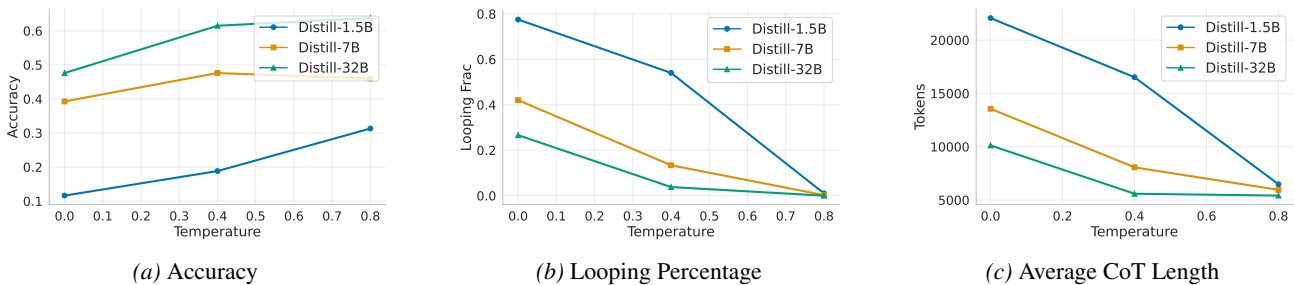

*(a)* Accuracy      *(b)* Looping Percentage      *(c)* Average CoT Length

*Figure 13.* Qwen metrics on GPQA as a function of temperature.

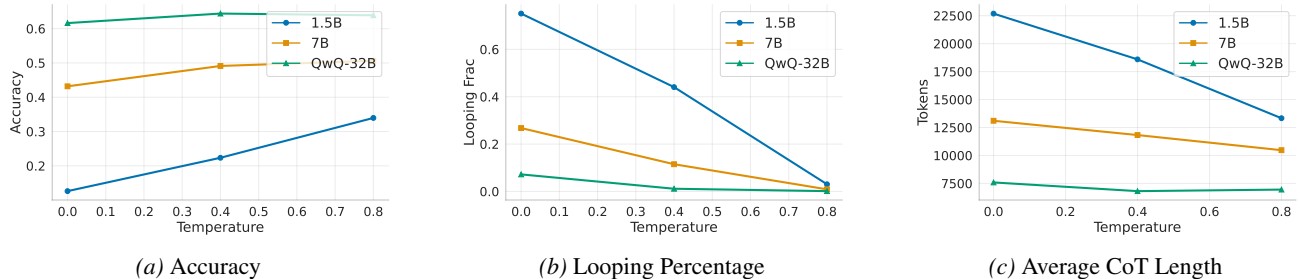

*(a)* Accuracy        *(b)* Looping Percentage        *(c)* Average CoT Length

*Figure 14.* Openthinker metrics on GPQA as a function of temperature.

## B. Example Traces of Openthinker-3 1.5B on AIME 2025 problems

### B.1. AIME 2025 Problem I-7 (Example 1)

The model is prompted to solve Problem I-7 from AIME 2025, with greedy decoding:

> The twelve letters $A$, $B$, $C$, $D$, $E$, $F$, $G$, $H$, $I$, $J$, $K$, and $L$ are randomly grouped into six pairs of letters. The two letters in each pair are placed next to each other in alphabetical order to form six two-letter words, and then those six words are listed alphabetically. For example, a possible result is $AB$, $CJ$, $DG$, $EK$, $FL$, $HI$. The probability that the last word listed contains $G$ is $m/n$, where $m$ and $n$ are relatively prime positive integers. Find $m + n$.

The model begins sensibly. It paraphrases the problem and correctly derives the total number of ways to pair up the letters.

> First, let me make sure I understand the problem correctly. We have the letters A, B, C, D, E, F, G, H, I, J, K, and L. These are to be randomly paired up into six pairs. Each pair is then arranged in alphabetical order to form a two-letter word. Then, all these six words are listed alphabetically. The question is asking for the probability that the last word in this list contains the letter G.
> . . .
> Let me start with the total number of ways to pair the letters. Since the letters are being paired into unordered pairs, the number of ways to partition 12 letters into 6 pairs is given by $(12)!$ divided by $(2^6 * 6!)$. . . .

Next, it tries to reason about how the two-letter words are formed and loops (semantically) for a bit, repeatedly going over how a word is arranged in alphabetical order.

> Now, the next part is figuring out how many of these pairings result in the last word (when the words are listed alphabetically) containing G. Hmm, so the key here is that the last word in the alphabetically sorted list must contain G. To do this, I need to think about how the words are formed and how their order is determined.
>
> Each pair is formed by taking two letters and putting them in alphabetical order. So, for any pair, say, (X, Y), if X comes before Y in the alphabet, then the word is XY; otherwise, it's YX. Wait, but actually, the problem says "the two letters in each pair are placed next to each other in alphabetical order to form six two-letter words". So, regardless of the order in the pair, they are arranged in alphabetical order. So, for any pair, the two letters are ordered such that the first is before the second in the alphabet. Therefore, each pair is effectively an ordered pair in the alphabetical order. So, for example, if the pair is (C, D), the word is CD. If the pair is (D, C), the word is DC, but since they are placed in alphabetical order, it's DC? Wait, no, actually, if the pair is (D, C), then since D comes after C, they would be placed as CD? Wait, no, hold on. Wait, the problem says "the two letters in each pair are placed next to each other in alphabetical order". So, regardless of the original order of the pair, they are arranged in alphabetical order. So, for any pair, the two letters are sorted, so the first letter is the earlier one, the second is the later one. So, for example, if the pair is (C, D), the word is CD. If the pair is (D, C), the word is DC. Wait, but actually, the pair is just two letters, so the pair is unordered, but when forming the word, they are arranged in alphabetical order. So, the word is always the two letters in order from A to Z. So, the word is the two letters sorted. So, the word is always the first letter being the earlier one, the second the later one. So, the word is the concatenation of the two letters in alphabetical order. So, for any pair, the word is the two letters in order. So, the word is determined by the two letters, but ordered alphabetically. . . .

After this, it briefly comes out of the loop but then slips into another one, now restating the goal of the problem without adding new structure.

> Alternatively, maybe it's better to think in terms of the last word in the sorted list. Since the words are sorted alphabetically, the last word is the largest one. So, the problem reduces to: what is the probability that the largest word in the list contains G?
>
> Therefore, the problem is equivalent to: when you partition the letters into six pairs, sort each pair into alphabetical order, then sort all the six words alphabetically, what is the probability that the last (i.e., the largest) word contains G.
>
> Hmm, okay. So, perhaps another way to think about this is: the last word is the maximum of the six words, so the maximum word must contain G. Therefore, the question is: what is the probability that the maximum word in the set of six sorted pairs is a word that contains G.
>
> . . . [repeated 2 more times]

Finally, it starts to argue about how pairs including G can be arranged, but again falls into a loop, repeating that words are formed by sorting the pairs, and never gets out of it.

> Alternatively, maybe it's better to think in terms of the possible pairs that include G and how they can be arranged.
>
> First, let me note that G is the 7th letter. So, the letters before G are A, B, C, D, E, F, and after G are H, I, J, K, L. So, G is in the middle. So, the pairs that include G can be either (G, something before G) or (something before G, G). But when sorted, the pair with G will be either (G, something before G) or (something before G, G) but sorted, so it will be (something before G, G). Wait, no. Wait, if the pair is (G, H), then the word is GH. If the pair is (H, G), the word is HG, but since they are sorted, it's GH. Wait, so actually, the word is always the two letters in alphabetical order, so the word is the pair sorted. Therefore, the word is always the two letters in order, so the word is the same regardless of the original pair. Therefore, the word is determined by the two letters, but sorted. Therefore, the word is the two letters in order. So, the word is the same for any pair of letters, but the pair is unordered. Therefore, the word is the two letters in order. So, the word is the concatenation of the two letters in alphabetical order. Therefore, the word is uniquely determined by the two letters, but the pair is unordered. Therefore, the word is the same for any pair of letters, but the pair is just the two letters. So, the key is that the word is the two letters sorted, so the word is the same as the pair sorted. Therefore, the word is the two letters in order, so the word is the same as the pair sorted. Therefore, the word is the two letters in order, so the word is the same as the pair sorted. Therefore, the word is the two letters in alphabetical order, so the word is the same as the pair sorted. Therefore, the word is the two letters in order, so the word is the same as the pair sorted. Therefore, the word is the two letters in order, so the word is the same as the pair sorted. Therefore, the word is the two letters in order, so the word is the same as the pair sorted. . . . [keeps repeating the exact same statement]

This trace illustrates a kind of risk-aversion: the model repeatedly takes easy cyclic actions (for example, repeating the same fact about how a two-letter word is formed, or restating the problem), instead of progress-making actions, such as exploring the structural insights needed to solve the problem. An example of structural insight is that for the last word to contain G, no pair should be formed with both letters coming from H-L.

**Further analysis.** To probe this behavior further, we truncate the trace at the point where the final looping begins (right after it says "Wait, so actually, the word is always the two letters in alphabetical order, so the word is the pair sorted.") and let the teacher model (QwQ-32B) continue from that prefix under greedy decoding. To account for non-determinism, we generate 5 continuations. We get 2 distinct traces: one reaches the correct answer, while the other misses a case and therefore undercounts. However, both traces avoid looping and make significant progress, capturing most of the structural insights. We also regenerate 5 completions from the student model from the same prefix and all of them get stuck in a loop.

To understand how much probability mass the student and teacher put on progress-making actions, we also sample 50 continuations of length 2000 tokens from each model at a high temperature. We then feed these continuations to GPT-5, provide it with a list of key structural insights, and ask whether each continuation contains at least one of them. At temperature 0.7, GPT-5 marks $1/50$ student continuations and $34/50$ teacher continuations as containing an insight (at temperature 1, these numbers are $8/50$ and $32/50$). Thus, the student puts much less mass on progress-making actions than the teacher.

Overall, in this example, we see that under greedy decoding the student demonstrates risk-aversion, preferring easy cyclic actions even when the teacher can make progress, and the teacher assigns substantially more probability mass to progress-making actions than the student.

### B.2. AIME 2025 Problem I-5 (Example 2)

Here, we prompt Openthinker-3 1.5B to solve Problem I-5 from AIME 2025, with greedy decoding:

> There are $8! = 40320$ eight-digit positive integers that use each of the digits $1, 2, 3, 4, 5, 6, 7, 8$ exactly once. Let $N$ be the number of these integers that are divisible by 22. Find the difference between $N$ and $2025$.

The model starts well. It recalls that being divisible by 22 means being divisible by both 2 and 11, notes that the last digit must be even, and sets up the standard 11-divisibility condition in terms of the sums of digits in odd and even positions.

> Okay, so I need to find the number of eight-digit numbers using each of the digits 1 through 8 exactly once that are divisible by 22. . . .
>
> First, a number divisible by 22 must be divisible by both 2 and 11. Since 22 is 2 times 11, and 2 and 11 are coprime, the number has to satisfy the divisibility rules for both 2 and 11.
>
> Starting with divisibility by 2: For a number to be divisible by 2, its last digit must be even. So the last digit has to be one of 2, 4, 6, or 8. That gives me a starting point.
>
> Now, divisibility by 11: The rule for 11 is that the difference between the sum of the digits in the odd positions and the sum of the digits in the even positions must be a multiple of 11 (including zero). Since we're dealing with an eight-digit number, the positions are 1 to 8. So, the first, third, fifth, seventh digits are the odd positions, and the second, fourth, sixth, eighth digits are the even positions.
>
> Let me formalize this. Let the number be D1 D2 D3 D4 D5 D6 D7 D8. Then, the sum of the odd positions is $S_{\text{odd}} = D1 + D3 + D5 + D7$, and the sum of the even positions is $S_{\text{even}} = D2 + D4 + D6 + D8$. The difference $(S_{\text{odd}} - S_{\text{even}})$ must be divisible by 11.

From here it makes correct algebraic progress on the 11-rule: it rewrites the condition in terms of $S_{\text{odd}}$ alone, reduces modulo 11, and finds that $S_{\text{odd}} \equiv 7 \pmod{11}$. It then uses the fact that $S_{\text{odd}}$ is the sum of four distinct digits from $\{1, \ldots, 8\}$ and correctly narrows down to the only feasible value.

Since all digits from 1 to 8 are used exactly once, the total sum of all digits is $1 + 2 + 3 + 4 + 5 + 6 + 7 + 8 = 36$. Therefore, $S_{\text{odd}} + S_{\text{even}} = 36$.

So, the difference $(S_{\text{odd}} - S_{\text{even}})$ must be a multiple of 11. Let me denote $D = S_{\text{odd}} - S_{\text{even}}$. Then, $D = S_{\text{odd}} - (36 - S_{\text{odd}}) = 2 * S_{\text{odd}} - 36$. Therefore, $2 * S_{\text{odd}} - 36$ must be divisible by 11.

So, $2 * S_{\text{odd}} \equiv 36 \pmod{11}$. Let me compute $36 \pmod{11}$. $11 * 3 = 33$, so $36 - 33 = 3$, so $36 \equiv 3 \pmod{11}$. Therefore, $2 * S_{\text{odd}} \equiv 3 \pmod{11}$.

To solve for $S_{\text{odd}}$, we can multiply both sides by the modular inverse of $2 \pmod{11}$. The inverse of $2 \pmod{11}$ is 6 because $2 * 6 = 12 \equiv 1 \pmod{11}$. Therefore, multiplying both sides by 6 gives $S_{\text{odd}} \equiv 3 * 6 \pmod{11}$, which is $18 \pmod{11}$, which is $7 \pmod{11}$.

Therefore, $S_{\text{odd}} \equiv 7 \pmod{11}$. Since $S_{\text{odd}}$ is the sum of four distinct digits from $1 - 8$, let's find all possible values of $S_{\text{odd}}$ that are congruent to $7 \pmod{11}$.

First, the possible range of $S_{\text{odd}}$. The minimal possible sum of four distinct digits is $1 + 2 + 3 + 4 = 10$, and the maximum is $5 + 6 + 7 + 8 = 26$. So $S_{\text{odd}}$ can be between 10 and 26.

We need $S_{\text{odd}} \equiv 7 \pmod{11}$. Let's see which numbers between 10 and 26 are congruent to $7 \pmod{11}$.

$7 \pmod{11}$ is 7, $7 + 11 = 18$, $18 + 11 = 29$, but 29 is beyond 26. So the only possible value is 18. Wait, $7 \pmod{11}$ is 7, but 7 is less than 10, so the next one is 18. 18 is within $10 - 26$, so $S_{\text{odd}}$ must be 18.

Wait, but $7 \pmod{11}$ is 7, but since $S_{\text{odd}}$ must be at least 10, the next possible is 18. So $S_{\text{odd}}$ must equal 18.

Therefore, the sum of the digits in the odd positions must be 18, and the sum of the even positions will then be $36 - 18 = 18$ as well.

At this point the model makes its first key mistake. When it fixes the last digit $E$ to be even and looks at the remaining 7 digits, it continues to insist that the four digits in the odd positions must sum to 18 and the three digits in the remaining even positions must also sum to 18, as if the last digit had disappeared.

Additionally, the last digit (position 8) must be even $(2, 4, 6, 8)$ to satisfy divisibility by 2.

Therefore, we can approach this problem in steps:

1. First, choose the last digit (position 8) to be even. There are 4 choices $(2, 4, 6, 8)$.

2. Then, partition the remaining 7 digits into two groups: the first, third, fifth, seventh positions (four digits) and the second, fourth, sixth positions (three digits). Wait, no, actually, the remaining digits after choosing the last digit are 7 digits, but the even positions (positions $2, 4, 6, 8$) need to have three digits (since position 8 is already chosen). Wait, let me clarify:

Wait, actually, the even positions are positions $2, 4, 6, 8$. Since position 8 is already fixed as an even digit, the remaining even positions (positions $2, 4, 6$) must be filled with three digits from the remaining seven digits (since one even digit is already used in position 8). The odd positions (positions $1, 3, 5, 7$) must be filled with the remaining four digits.

Therefore, the total digits to assign are 7 digits (since one even digit is used in position 8). These 7 digits must be split into two groups: four digits for the odd positions (positions $1, 3, 5, 7$) and three digits for the even positions (positions $2, 4, 6$). The sum of the four digits in the odd positions must be 18, and the sum of the three digits in the even positions must be 18 (since total sum is 36, so $36 - 18 = 18$).

This leads to a contradiction ($E$ would have to be 0), and the model notices that something is wrong.

Therefore, the problem reduces to:

...
First, let me note that the last digit is fixed as an even digit $(2, 4, 6, 8)$. Let's denote the last digit as $E$.

...
We need to choose 4 digits from these 7 to be in the odd positions (positions 1,3,5,7) such that their sum is 18, and the remaining 3 digits will be in the even positions (positions 2,4,6) with sum 18.
Therefore, for each E, we need to compute the number of 4-element subsets of the remaining 7 digits that sum to 18, and then for each such subset, the number of permutations is 4! (for the odd positions) multiplied by 3! (for the even positions).

...
Wait, let's see. Let me first compute the total sum of the remaining 7 digits when E is removed. The total sum of all digits is 36, so the remaining sum is $36 - E$.

...
Therefore, the sum of the 3 digits (even positions) is 18, so the sum of the 4 digits (odd positions) is 18.

...
Therefore, the total sum of the 7 digits (excluding E) is $18 + 18 = 36 - E$.
Therefore, $36 - E$ must equal 36, so that 36 - E = 36? Wait, that can't be. Wait, $18 + 18 = 36$, so $36 - E$ must equal 36? That would mean $E = 0$, but $E$ is an even digit from $2, 4, 6, 8$.

...
Wait, that's a contradiction. That suggests that my earlier reasoning is wrong.
Hmm, so where did I go wrong? Let me check again.

However, instead of changing its approach, it falls into a loop: it repeatedly restates the divisibility-by-11 condition, re-derives $S_{\text{odd}} \equiv 7 \pmod{11}$, concludes again that $S_{\text{odd}} = 18$, and runs back into the same contradiction with $E$.

> Wait, let me go back.
> We had $S_{\text{odd}} - S_{\text{even}} \equiv 0 \pmod{11}$, and $S_{\text{odd}} + S_{\text{even}} = 36$.
>
> ...
>
> Therefore, $S_{\text{odd}} \equiv 7 \pmod{11}$.
>
> ...
>
> But then, the total of the seven digits (excluding the last digit E) must be $18 + 18 = 36$, so $36 - E = 36 \Rightarrow E = 0$.
> But E is an even digit from 2,4,6,8.
> This is a contradiction.
> [Semantically loops essentially repeating the same mistake 5 more times]

This behavior bears similarity to the temporally correlated errors we saw in our graph reasoning setting: once the model makes a high-level mistake, it keeps returning to the same wrong reasoning whenever it tries to restart. In the graph setting, this corresponded to repeatedly picking the same wrong child at the root, leading to the same leaf, and restarting.

After several rounds of semantic looping, the model makes a new mistake! It now claims that $S_{\text{odd}} + S_{\text{even}} = 36 - E$ instead of 36, which it had correctly used earlier. Building on this, it mistakenly concludes $2S_{\text{odd}} \equiv 36 - E \pmod{11}$ (instead of $2S_{\text{odd}} \equiv 36 \pmod{11}$, which it correctly found earlier). It follows up with a case analysis over $E \in \{2, 4, 6, 8\}$.

> Wait, $S_{\text{odd}} + S_{\text{even}} = 36 - E$.
>
> ...
>
> Therefore, $2S_{\text{odd}} \equiv 36 - E \pmod{11}$.
>
> Ah! So my earlier mistake was not including the E term.
>
> ...
>
> Therefore, for each $E$, possible $S_{\text{odd}}$ values are:
>
> $E = 2$: $S_{\text{odd}} = 17$
> $E = 4$: $S_{\text{odd}} = 16$
> $E = 6$: $S_{\text{odd}} = 15$ or $26$
> $E = 8$: $S_{\text{odd}} = 14$ or $25$

From here, the model switches to a more combinatorial counting approach: for each fixed even last digit $E$, it wants to count the number of 4-element subsets of the remaining 7 digits whose sum is the required $S_{\text{odd}}$, and then multiply by $4! \cdot 3!$ to account for permutations in the odd and even positions.

> Therefore, now, for each $E$, we can compute the number of 4-element subsets of the remaining 7 digits that sum to the required $S_{\text{odd}}$.
> Then, for each such subset, the number of permutations is $4! * 3!$.
> Therefore, the total $N$ is the sum over each $E$ of [number of subsets for $E$] $* 4! * 3!$.
>
> ...
>
> Starting with $E = 2$:
> Remaining digits: $1, 3, 4, 5, 6, 7, 8$.
> Need subsets of 4 digits that sum to 17.
> Let me list all possible 4-element subsets and see which sum to 17.
> Alternatively, perhaps a smarter way.
> The total sum of the remaining digits is $1 + 3 + 4 + 5 + 6 + 7 + 8 = 36 - 2 = 34$.
> We need subsets of 4 digits summing to 17, so the remaining 3 digits will sum to $34 - 17 = 17$.
> Therefore, the problem reduces to finding the number of 4-element subsets of the 7 digits that sum to 17, which is equivalent to the number of 3-element subsets that also sum to 17.
> Therefore, the number of such subsets is equal to the number of 3-element subsets of $\{1, 3, 4, 5, 6, 7, 8\}$ that sum to 17.

When it actually tries to carry this out (starting with $E = 2$), it gets stuck in a second loop. Instead of systematically listing and counting the subsets, it repeatedly writes down the same few candidate triples and partial checks, without moving on to finish the enumeration or to the other values of $E$.

Let me compute that.

Start with the largest digit, 8:
- $8 + 7 + 2 =$ but 2 is not in the set.
Wait, digits are $1, 3, 4, 5, 6, 7, 8$.
Looking for triples adding to 17.
Start with 8:
$8 + 7 + 2 =$ but 2 not there.
$8 + 6 + 3 = 17$.
Yes, $8 + 6 + 3 = 17$.
$8 + 5 + 4 = 17$.
$8 + 7 + 2 =$ no.
$8 + 5 + 4 = 17$.
$8 + 6 + 3 = 17$.
Next, $8 + 7 + 2$ invalid.

...

Wait, $8 + 5 + 4 = 17, 8 + 6 + 3 = 17, 8 + 7 + 2 =$ no.
Wait, $8 + 5 + 4 = 17, 8 + 6 + 3 = 17, 8 + 7 + 2 =$ no.
Wait, $8 + 5 + 4 = 17, 8 + 6 + 3 = 17, 8 + 7 + 2 =$ no.

[Loops for a bit repeating the same line.]
...

Wait, let's try without 8.

The maximum sum without 8 is $7 + 6 + 5 = 18$, which is more than 17.
So possible combinations:
$7 + 6 + 4 = 17$.
$7 + 5 + 5 =$ no.
$7 + 6 + 4 = 17$.
$7 + 5 + 5 =$ no.
$7 + 6 + 4 = 17, 7 + 5 + 5 =$ no.
$7 + 6 + 4 = 17, 7 + 5 + 5 =$ no.
$7 + 6 + 4 = 17, 7 + 5 + 5 =$ no.
Wait, $7 + 6 + 4 = 17, 7 + 5 + 5 =$ no.
Wait, $7 + 6 + 4 = 17, 7 + 5 + 5 =$ no.
Wait, $7 + 6 + 4 = 17, 7 + 5 + 5 =$ no.
Wait, $7 + 6 + 4 = 17, 7 + 5 + 5 =$ no.
Wait, $7 + 6 + 4 = 17, 7 + 5 + 5 =$ no.

[Gets stuck here repeating the same line and never gets out.]

This second loop demonstrates a risk-aversion pattern. By this point, the model has already identified the three valid triples in the $E = 2$ case. The natural progress-making step would be to conclude that there are three such triples, convert this into a count of permutations for $E = 2$, and then move on to the next value of $E$. Instead, it keeps revisiting the same triples again and again, rechecking them rather than advancing the argument.

**Further analysis (first loop).** To probe this behavior further, we truncate the trace at the point where the model first reaches a contradiction (right after it concludes that $E$ would have to be 0). From this prefix, we let the teacher model (QwQ-32B) continue under greedy decoding and generate 5 continuations to account for non-determinism. In all traces, the teacher immediately identifies the mistake, repairs the reasoning, and eventually reaches the correct answer. We also generate 5 new continuations from the student (OpenThinker-3 1.5B) from the same prefix. This time we see two distinct student traces: both loop 2-3 more times, repeatedly re-deriving the same wrong condition and contradiction, but eventually notice the mistake. Once they exit this first loop, they start a case-by-case analysis in $E$ and then get stuck again in a loop, similar to the second loop in the above trace.

To get a quantitative sense of how much probability mass the two models place on the progress-making action of correctly escaping the contradiction, we sample 50 continuations of length 1000 tokens from each model at temperatures 0.7 and 1.0 from the same truncated prefix. We then ask GPT-5 to check whether each continuation correctly fixes the mistaken equation and moves on with valid reasoning. At temperature 0.7, GPT-5 marks $19/50$ student continuations and $50/50$ teacher continuations as successfully escaping the contradiction; at temperature 1.0, these numbers are $21/50$ and $48/50$, respectively. Thus, the teacher puts much more mass on the correct progress-making action than the student.

Overall, this first loop shows glimpses of the temporally correlated errors mechanism from Section 4: once the student falls into a wrong pattern, it tends to revisit the same mistaken step several times. Here, the gap between the teacher and the student is relatively large, for example in terms of how much probability mass they put on the correct progress-making action. In this sense, the behavior is closer in spirit to our margin variant (where, after pursuing an action once, the teacher

shifts more probability to other actions) or to the hardness+correlated-errors variant in Appendix E, where we show that temporally correlated errors can arise even when the teacher distribution puts high mass on the correct progress-making action.

**Second loop.** We also probe the second looping point by truncating the trace right when the model gets stuck in the final infinite loop (at the point after it says "$7 + 6 + 4 = 17, 7 + 5 + 5 = $ no"). From this prefix, we again generate 5 greedy continuations from both the teacher (QwQ-32B) and the student (OpenThinker-3 1.5B). Interestingly, in this case, along with the student, all teacher continuations gets stuck in the same loop.

Further, we sample 50 continuations of length 2000 tokens from each model at temperatures 0.7 and 1.0, starting from the same truncated prefix. We then ask GPT-5 whether each continuation makes progress, where we define progress as continuing the enumeration and correctly counting the valid tuples for the $E = 2$ case. At temperature 0.7, GPT-5 marks 10/50 student continuations and 34/50 teacher continuations as making progress; at temperature 1.0, these numbers are 27/50 and 42/50, respectively. Thus, the teacher still puts more mass on the correct progress-making actions than the student, but the gap is less substantial, especially at temperature 1.

This second loop is interesting because here we see risk-aversion–style looping in both the student and the teacher. There are several possible reasons for why a teacher might also loop.

First, the teacher itself may carry errors in learning from its own training process. Even if it was trained with reinforcement learning, hardness of learning (as discussed in Section 3.3) can still cause it to loop.

Second, note that the teacher did not loop when we started it from the earlier looping point: from that earlier prefix, it was able to identify the mistake and reach the correct answer. In contrast, when we start it from the later point, it is conditioned on a student trace that has already semantically looped several times (repeating the same mistaken reasoning and contradiction). This suggests that error accumulation in the prefix may also play a role: for example, in-context learning from a "confused" history could cause the teacher itself to behave in a more confused, looping way. Understanding the precise mechanisms by which this kind of error accumulation drives loops is an interesting direction for future work. Past work has also noted that looping sequences are often preceded by less natural text, which is consistent with error accumulation playing a role (Chiang & Chen, 2021) .

## C. Proof of Proposition 3.1 and mapping to language models

We restate the proposition below and provide its proof.

**Proposition C.1.** *Consider the following task: there exist $n$ sets of contexts $C_1, \ldots, C_n$ which are equi-likely under the training distribution. And there are $n$ distinct "hard" actions $a_1, \ldots, a_n$, and an "easy" action $a_0$. For every context $c_i \in C_i$, the training distribution picks action $a_i$ with probability $(1 - p)$ and $a_0$ with probability $p$. Now consider a learner that cannot distinguish between the $n$ hard actions or, in other words, it is constrained to ignore the context when deciding on the best action. Then the maximum log-likelihood solution for such a learner assigns probability $p$ to the easy action $a_0$ and probability $(1 - p)/n$ to the hard indistinguishable actions $a_i, \forall i \in \{1, \ldots, n\}$.*

*Proof.* Let $q_1, \ldots, q_n$ be the probabilities that the model assigns to the $n$ actions, and $q_0$ be the probability of the easy action. The Cross-Entropy loss for a context $c_i \in C_i$ with correct action $a_i$ is

$$\ell_i = -p \log(q_0) - (1 - p) \log(q_i).$$

Since all $n$ contexts are equi-probable, the average Cross Entropy over the dataset will be

$$\ell = \frac{1}{n} \sum_{i=1}^{n} \ell_i$$

$$= -p \log(q_0) - (1 - p) \frac{1}{n} \sum_{i=1}^{n} \log(q_i).$$

Note that $\sum_{i=0}^{n} q_i = 1$ and $q_i \geq 0$ for all $i$, by definition. We can further assume that $q_i > 0$, otherwise the loss would be infinite. This is a convex minimization problem under linear constraints, so to find the minimizer it suffices to look at the points satisfying the KKT conditions. In particular,

$$\frac{p}{q_0} = \mu \implies q_0 = \frac{1}{\mu} p$$

$$\frac{1 - p}{n} \frac{1}{q_i} = \mu \implies q_i = \frac{1}{\mu} (1 - p)/n \ \text{ for } i = 1, \ldots, n,$$

for some $\mu \in \mathbb{R}$. Using the equality constraint, we conclude that $\mu = 1$ and $q_0 = p$, $q_i = (1 - p)/n$ for $i = 1, \ldots, n$ is the unique minimizer of the problem, as desired. $\qquad\square$

**Mapping to language models.** In a language model, an action can be viewed as a short chunk of tokens implementing a logical step (e.g., the next step in a proof). For a span $x_{t:t+k-1}$, the model's probability of that action given the prefix is $P_\theta(x_{t:t+k-1} \mid x_{<t}) = \prod_{i=t}^{t+k-1} P_\theta(x_i \mid x_{<i})$, so the log-probability of the chunk is the sum of the per-token log-probabilities. Because next-token training maximizes this sum over tokens, it also maximizes the log-probability of any such chunk. Consequently, if a progress-making step is hard (confusable with $n$ alternative chunks), the model spreads its mass across those chunks, reducing the learned probability on the intended chunk by a $1/n$ factor, while an easy cyclic span retains its mass. The proposition therefore applies directly to these chunks.

## D. Risk-aversion due to Hardness of Learning: Further details and implications

### D.1. Training Details

We train a decoder-only Transformer from scratch, with 12 layers, 8 attention heads, and 768 embedding dimension ($\approx 85M$ parameters). We use Adam for 100k steps with a learning rate of $10^{-4}$ (cosine decay) and batch size 64. We use 2M training sequences and train with cross-entropy loss for next-token prediction, applying the loss only to the random-walk portion of the sequence. At test time, the model receives new randomly generated instances and, given the edge list, start, and goal, is expected to generate a walk from the start to the goal. We mark a generated walk as accurate if it takes only valid transitions and eventually reaches the goal and stops.

### D.2. More implications

**Instruct vs Reasoning Models.** Two ingredients are needed in the mechanism above: (i) hard-to-learn actions, along with (ii) easy-to-learn cyclic actions present in the training distribution. The presence of hard actions amplifies the frequency of easy cyclic actions in model generations. However, if cyclic actions are rare in the training traces, extensive looping is less likely. This is a plausible explanation for why many instruction-tuned models loop less than reasoning models as reasoning traces include more cyclic actions such as re-statement and backtracking.

**RL vs distillation.** The mechanism discussed in this section can affect models trained via RL too, and not just distillation. A guiding principle here is that whenever there is a capacity difference between the teacher model and the student model, hardness of learning for the student can amplify looping behavior. One can approximately view the RL training process as sampling multiple trajectories from the model and training on the correct ones. While there is no explicit teacher model for a RL trained model, this can be thought of as training the model on best-of-$k$ version of itself where $k$ is the number of trajectories sampled. In that sense, there is still a gap between the data generator and the learner, which can possibly cause this mechanism.

## E. Looping via a combination of Hardness + Correlated Errors

In Section 3, we saw how hardness of learning can cause looping and in Section 4 we saw how an inductive bias for correlated errors can lead to looping. In this section, we present yet another variant showing how both hardness and correlated errors can act together and cause looping. In particular, we show that Transformers can have temporally correlated errors even when the teacher puts substantial mass on the correct progress-making action, as opposed to the random uniform distribution we considered in Section 4.

**Setup.** We use the same star graph and random walk as in Section 4, with a minor modification at the root. Instead of choosing a child uniformly at random at the root, the training distribution now goes to the correct child (leading to goal) with probability 0.5 and with probability 0.5, chooses one of the other children uniformly at random. We train a Transformer on traces drawn from this process for the $G(5, 5)$ graph; training and test details match the previous synthetic experiments.

For our looping metric, like in Section 4, we use the maximum count of times that a root child is visited and average over all 4000 generated traces. The results for $G(5, 5)$ are shown in Figure 15. We observe non-trivial amount of looping at temperature 0, but the looping is not as strong as the walk that picked uniformly random children at the root. Note that a perfect learner would directly go to the correct child with greedy decoding, which would correspond to a looping count of 1.

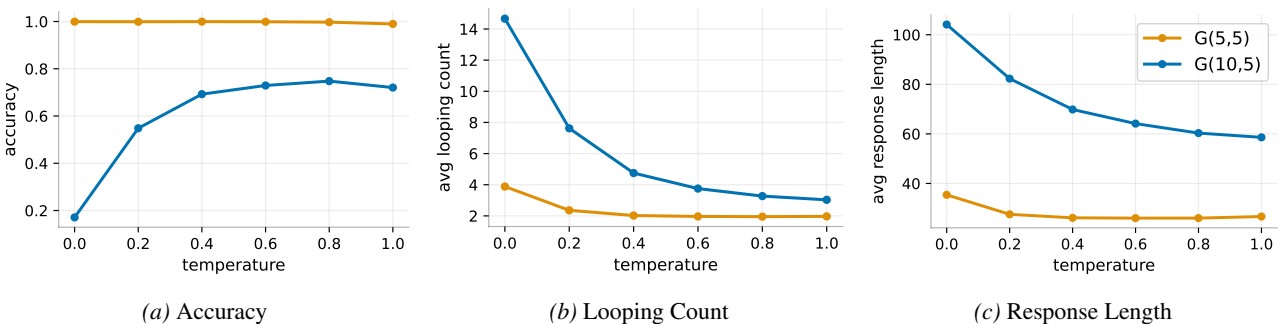

*(a)* Accuracy      *(b)* Looping Count      *(c)* Response Length

*Figure 15.* **Looping from combined hardness and temporally correlated errors.** We work in a variant of the correlated-errors setting from Section 4. For $G(5, 5)$ (orange), the setup is the same as in the previous section except for a minor change at the root: instead of choosing a child uniformly, the training distribution now goes to the correct child with probability 0.5 and to each of the four wrong children with probability $0.5/4$. For $G(10, 5)$ (blue), the root distribution again puts 0.5 on the correct child but spreads the remaining 0.5 over only four of the nine wrong children. On $G(5, 5)$, looping is still present but relatively weak and accuracy is near perfect. On $G(10, 5)$, looping is much stronger at low temperature and response lengths are substantially longer, especially under greedy decoding. Overall, this shows that looping similar to Section 4 can occur even when the training distribution places significant mass on the correct progress-making action.

**Interpretation.** Similar to Section 3, the model initially finds it hard to distinguish between the actions available at the root. This hardness of learning pushes it toward placing roughly equal mass on all root children, even though the training distribution gives probability 0.5 to the correct child and $0.5/4$ to each of the four incorrect children. Once this happens, the situation seems similar to the correlated-errors setting: each time the model reaches the root, it behaves almost as if it is

choosing a child uniformly at random, and temporally correlated errors can cause it to revisit the same children and loop.

To understand why looping is weaker here than in Section 4, it is helpful to condition on the model having already visited one wrong child, gone to its leaf, and then returned to the root. At this point, the training distribution still assigns probability $0.5$ to the correct child and $0.5/4$ to each wrong child, but a reasonable learner can now at least tell that the previously visited child is bad. A simple way to model this is to assume that the learner can give the visited child its own probability, while treating the remaining four children (three unseen wrong children plus the correct child) as indistinguishable. Under this constraint, the maximum-likelihood solution puts $0.5/4 = 0.125$ on the already-visited child and spreads the remaining $1 - 0.125 = 0.875$ uniformly over the other four children, giving about $0.219$ probability to each. In other words, compared to a perfectly uniform $1/5 = 0.2$ split, the visited child is slightly downweighted, which is very similar to the margin variant from Section 4 with a margin of about $0.075$. On top of this effective distribution, the Transformer's temporally correlated errors once again bias it toward reselecting a few children. Indeed, the looping curves in Figure 15 look very similar to those for margin $m = 0.1$ in Section 4. We also observe non-trivial temporal correlation: across consecutive root visits under greedy decoding, the top (and second) root child is the same $58.8\%$ ($47.5\%$) of the time, placing this variant between the $m=0.05$ ($75.3\%/52.7\%$) and $m=0.1$ ($52.5\%/40.1\%$) variants from Section 4.

**Inducing more severe looping.** The explanation above also suggests how to induce more severe looping by making the task harder for the student. Consider a $G(10, 5)$ graph. At the root, the teacher places probability $0.5$ on the correct child and spreads the remaining $0.5$ uniformly over four incorrect children, giving probability $0.5/4$ to each of these, while never visiting the remaining 5 children. Thus the teacher only ever explores five root children in total. Now condition again on the model having visited one of the wrong children once and returned to the root. A learner that can recognize this visited child as bad, but still cannot distinguish the remaining nine children from one another, will put about $0.5/4 \approx 0.125$ probability on the visited child and spread the remaining $1 - 0.125 = 0.875$ uniformly over the other nine children, giving roughly $(1 - 0.5/4)/9 \approx 0.097$ to each. Under greedy decoding, the model will then prefer the already-visited child at the root and keep returning to it, producing stronger loops. We implemented this variant and include its results in Figure 15; as we expected, it exhibits substantially more looping. We also observe substantial temporal correlation: across consecutive root visits under greedy decoding, the top (and second) root child is the same $92.0\%$ ($58.0\%$) of the time.

Overall, these instances show that looping similar to Section 4 can occur even when the teacher distribution places significant mass on the correct progress-making action. They also illustrate that the basic principles in this paper can combine in different ways to cause looping; a problem need not match exactly the setups in Sections 3 and 4 for these mechanisms to appear.

## F. Further Details on the Catalyst for Looping

This appendix provides further details on the catalyst discussed in Section 5. We first show confidence buildup in real model traces (Figure 16), then give a more detailed overview of what happens under the risk-aversion mechanism in Figure 17 including panel (b) for when there is no exploration, and finally demonstrate the catalyst effect in the correlated errors mechanism of Section 4.

Figure 16 illustrates this effect. We take the two looping traces discussed in Section 2.1 (AIME 2025 Problem I-7 and Problem I-5, generated with greedy decoding using the OpenThinker3-1.5B model; also discussed in Examples B.1 and B.2). For each trace, we plot the highest probability assigned to any next token at each decoding step. The red line marks the step where the final infinite loop begins. As the loop progresses, the maximum probability rises, showing that the model becomes increasingly confident in continuing the loop.

### F.1. Further details: risk-aversion mechanism.

We now reproduce the catalyst effect in the risk-aversion setup of Section 3. As also discussed in Section 5, we modify the training distribution so that with probability 0.001 the training example is a trace that deterministically loops between the start and root, and otherwise it is sampled as before. We report results for $G(5, 5)$ both with exploration (panel (a) of Figure 17, also shown in the main text) and without (panel (b)). For 4000 model-generated traces with greedy decoding, we plot the probability assigned to the reset (start) action each time the model visits the root. Without the bias, this probability stays around its learned value near the teacher's 0.3. When we add the bias, the reset probability rises toward 1 for most traces over successive visits, showing that the model becomes nearly certain it should keep looping once the loop has gone on for a while.

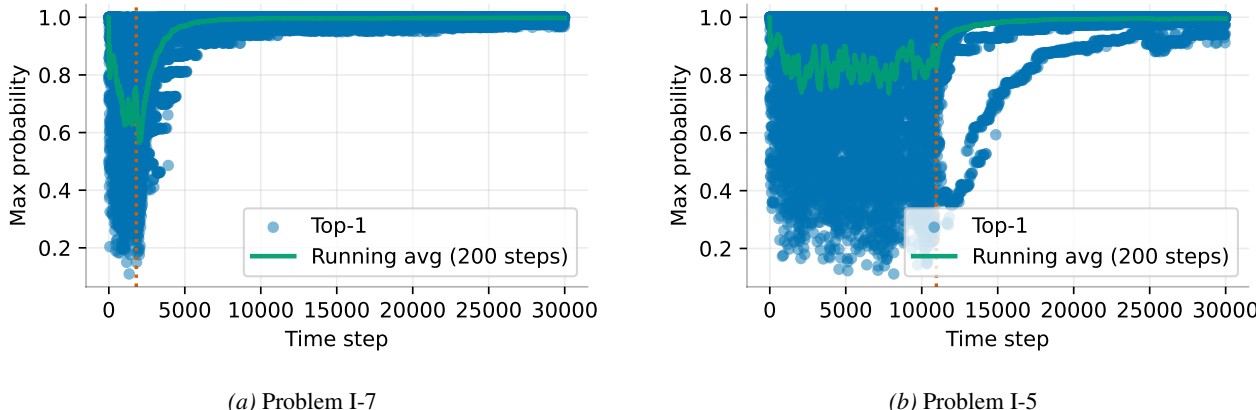

*(a)* Problem I-7                                    *(b)* Problem I-5

*Figure 16.* **Confidence buildup during looping.** We plot the maximum next-token probability over time for two looping traces from OPENTHINKER3-1.5B on AIME 2025 Problem I-7 (a) and Problem I-5 (b), generated with greedy decoding. Each blue dot shows the top-1 probability at a single decoding step; the apparent vertical bands are due to plotting 30k time steps at this resolution. The green curve is a running average (window size 200). The vertical dashed line marks the step where the final infinite loop begins. In both cases, once the loop starts, the top-1 probability rises toward 1, indicating that the model becomes increasingly confident in continuing the loop.

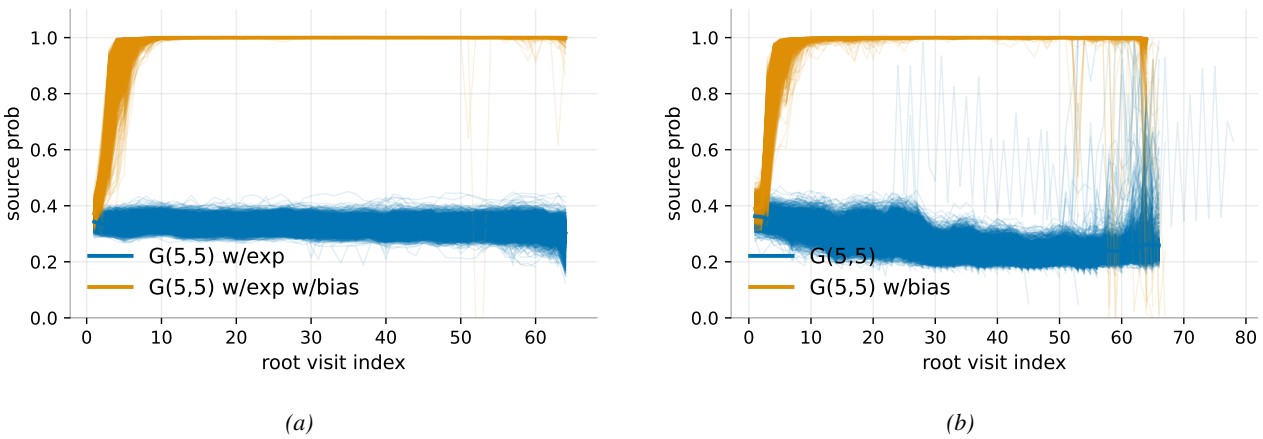

*(a)*                                    *(b)*

*Figure 17.* **Explicit looping bias drives reset probability toward 1.** We work in the risk-aversion setting on $G(5, 5)$ (Section 3), with and without exploration. For each of 4000 greedy-decoded traces, we record at every visit to the root the model's probability of taking the reset-to-start action ("source prob") and plot it against the root-visit index (thin lines). The thicker lines show the average over all traces that reach that visit index (traces that have already ended are not included in the average; the average is shown only at visit indices reached by at least 5% of traces). In the biased models (orange), the training distribution is modified so that with probability 0.001 the example is a trace that deterministically loops between start and root, and with the remaining probability it is sampled as before. Under this bias, the reset probability rises toward 1 for most traces as the number of visits grows, so once a loop has persisted for a while the model becomes almost certain to continue it. In the unbiased models (blue), the reset probability stays roughly constant in the exploration variant (a) and generally drifts downward in the non-exploration variant (b). Note that panel (a) also appears in the main text in Figure 5.

Figure 18 reports accuracy, looping counts, and response lengths versus temperature for $G(5,5)$ with and without exploration (similar to Section 3), comparing models trained with and without the added bias. With the bias, looping is clearly amplified. For example, on $G(5,5)$ without exploration, the original model typically loops between the start and root for a while and then escapes to some root child, reaching an arbitrary leaf; this yields accuracy close to chance under greedy decoding. After adding the bias, once the model has looped for some time, the reinforced reset probability keeps it trapped between the start and the root, and greedy-decoding accuracy drops close to zero.

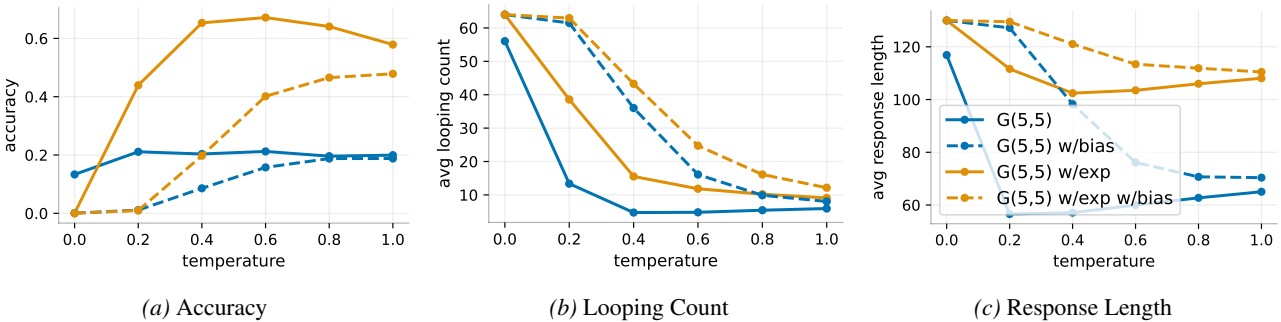

*(a)* Accuracy        *(b)* Looping Count        *(c)* Response Length

*Figure 18.* **Confidence bias amplifies looping and degrades accuracy.** In the risk-aversion setting on $G(5,5)$ (Section 3), we plot accuracy, looping count, and response length versus temperature, with and without exploration, comparing models trained on the original distribution (solid) and with an added looping bias (dashed). The bias increases low-temperature looping counts and response lengths and, in the non-exploration setting, drives greedy-decoding accuracy close to zero by keeping the model trapped between the start and root once a loop has formed.

### F.2. Correlated errors mechanism.

Next we revisit the correlated errors mechanism from Section 4. Here again we introduce a tiny looping bias. With a small probability 0.001, the training distribution picks a random non-goal leading child at the root (fixed once at the start of the trace) and generates a loop: from the root it goes to that child, walks forward to the leaf, resets to the start, returns to the root, takes the same child again, and so on. With the remaining probability, traces are drawn as before.

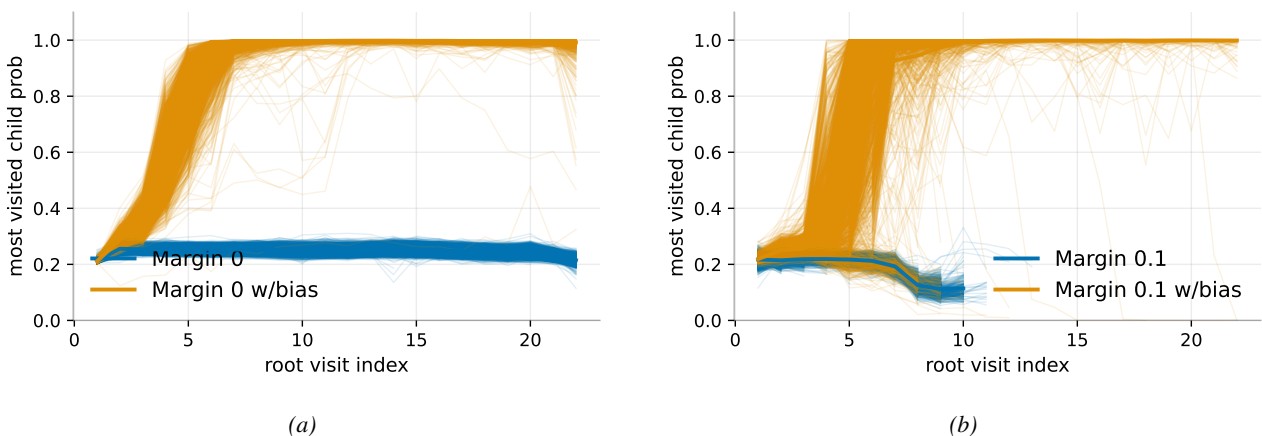

*(a)*                      *(b)*

*Figure 19.* **Explicit looping bias reinforces a single child at the root.** For the correlated-errors setting on $G(5,5)$, we compare models trained with and without an additional looping bias, for margin 0 (a) and margin 0.1 (b). The bias is introduced by modifying the training distribution so that, with small probability 0.001, a trace repeatedly goes from the root to a fixed non-goal child, walks to its leaf, resets, and repeats. For each greedy-decoded trace, we identify the most visited child at the root and, at every visit to the root, record the probability assigned to that child ("most visited child prob"). Thin lines show individual traces; thicker lines show the average over traces that reach each visit index (shown only at visit indices reached by at least 5% of traces). With the added bias (orange), this probability rises toward 1 for most traces as the number of visits grows. In the margin-0.1 setting, some traces leave the loop early, but most traces that loop for a bit become increasingly confident in choosing the same child again. Without the bias (blue), the probability on the most visited child stays roughly flat or drifts downward.

As before, this small change amplifies looping. In Figure 19, for each trace we first identify the most visited child at the

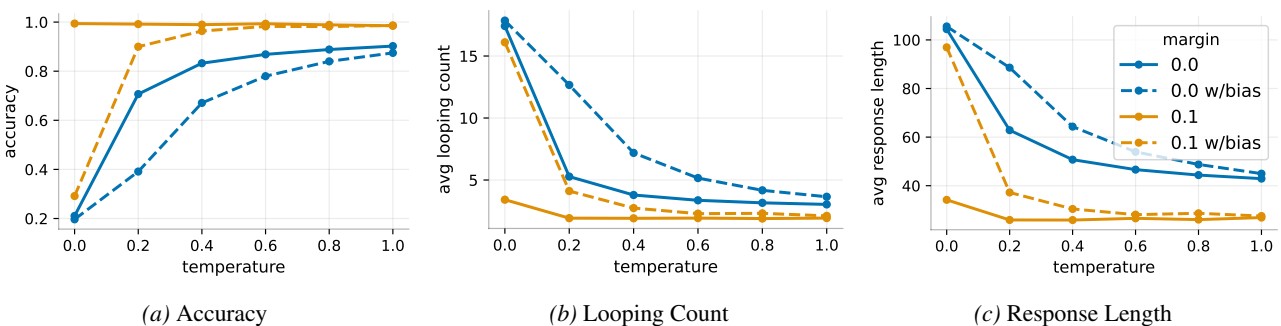

*(a)* Accuracy        *(b)* Looping Count        *(c)* Response Length

*Figure 20.* **Confidence bias amplifies looping under temporally correlated errors.** In the correlated-errors setting on $G(5,5)$ (Section 4), we plot accuracy, looping count, and response length versus temperature for margin 0 and margin 0.1, with and without an added looping bias. Solid lines use the original training distribution; dashed lines add the bias that occasionally generates traces which always revisit the same non-goal child at the root. The bias increases low-temperature looping counts and response lengths, and decreases low temperature accuracy. The effect is especially significant for margin 0.1 setting.

root, and then plot the probability assigned to that child at each visit to the root. For the margin 0 variant (panel (a)), this probability steadily increases and moves close to 1 when the bias is present. For the margin 0.1 variant (panel (b)), some trajectories leave the loop early, but for those that keep looping for a while, the probability on the most-visited child is again reinforced and rises toward 1.

Figure 20 shows accuracy, looping count, and response length versus temperature, similar to Section 4. Here too we see amplified looping. In particular, for the margin 0.1 setting, we previously observed only mild looping; with the added bias, looping becomes much stronger and accuracy drops noticeably, especially at low temperatures.

## G. Model capacity as a source of hardness

To complement Section 3 and isolate the effect of model capacity on looping, we train models of three different sizes on the $G(5,3)$ task (no exploration variant): a 1-layer model (7.1M parameters), a 2-layer model (14.2M parameters), and a 12-layer model (85M parameters; the same architecture used throughout the rest of the paper). All other hyperparameters are identical to those in Appendix D. To further increase the difficulty for the smaller models, we also increase max_nodes, the size of the node-label vocabulary that each random walk is drawn from, from 30 (its value in our other experiments) to 50 (we do this increase for all 3 models discussed in this section). Results across temperatures are shown in Figure 21.

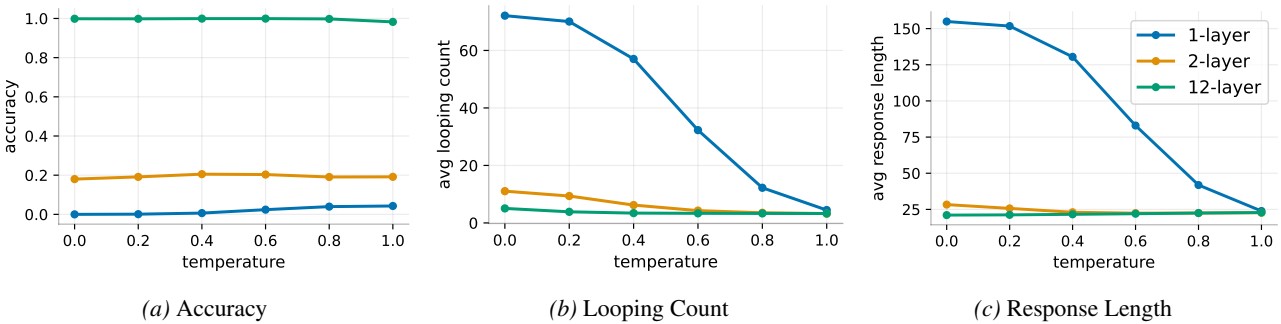

*(a)* Accuracy        *(b)* Looping Count        *(c)* Response Length

*Figure 21.* **Model capacity affects looping on $G(5,3)$.** We train 1-layer, 2-layer, and 12-layer models on the $G(5,3)$ task (no exploration variant) from Section 3, and plot accuracy, looping count, and response length against temperature. The 12-layer model achieves near-perfect accuracy across temperatures; the 2-layer model loops more than the 12-layer model at low temperature and plateaus at chance accuracy; the 1-layer model loops even more and plateaus below chance because it samples invalid edges at higher temperatures.

The 12-layer model reaches near-perfect accuracy at all temperatures and barely loops, matching its behavior on $G(5,3)$ in Section 3. The 2-layer model is affected by reduced capacity: at $T{=}0$ it loops on average about 11 times between source and root, compared to $\approx 5$ for the 12-layer model. After looping for some time, it commits to an arbitrary root child and walks to its leaf, yielding roughly chance accuracy. This mirrors the behavior we observed for $G(5,5)$ without exploration in Section 3, although the source of hardness there was optimization rather than capacity. The 1-layer model fares worse.

At $T{=}0$ it loops on average about 72 times between source and root, far more than either the 2-layer or 12-layer model. Its accuracy at $T{=}0$ is $0\%$ and reaches only $\approx 4\%$ at $T{=}1$, well below the $20\%$ chance baseline. Inspection of its traces shows that at higher temperatures it frequently samples tokens corresponding to invalid edges in the graph, which caps its achievable accuracy. In both small models, when the walk reaches the root, the model places higher probability on the reset action while diffusing the remaining mass across the root children, mirroring the diffused-mass pattern that drives the risk-aversion mechanism in Section 3.

We chose $G(5, 3)$ for this ablation deliberately. On $G(5, 5)$ the source of hardness is optimization: even a large 12-layer model cannot reliably learn the correct first edge from the root, as the problem reduces to learning a parity-like instance (Bachmann & Nagarajan, 2024; Hu et al., 2025), and model capacity is not the bottleneck. On $G(5, 3)$ the optimization hardness goes away (our 12-layer model solves the task near-perfectly), so $G(5, 3)$ is the natural setting to isolate model capacity as an independent source of hardness. The fact that shrinking the model from 85M to 14.2M and 7.1M parameters reintroduces severe looping confirms that capacity-driven hardness produces risk-aversion behavior similar to that of optimization-driven hardness, discussed in Section 3.

