# OpenReview forum: "Wait, Wait, Wait... Why Do Reasoning Models Loop?"
_ICML.cc/2026/Conference — ICML 2026 spotlight_

### Official Review · Reviewer_cMjw · 2026-02-20

**Soundness:** 2
**Presentation:** 3
**Significance:** 2
**Originality:** 2
**Overall Recommendation:** 4
**Confidence:** 2

**Summary:**

Testing on AIME and GPQA datasets, this paper finds that under low temperatures, looping is very common in reasoning models. Also, distilled students loop more than their teacher. These findings point to errors in learning. Then, this paper uses a star graph reasoning task to interpret how such errors lead to loops. There are two main reasons: risk-aversion and an inductive bias toward temporally correlated errors. Through the transformer training on graph tasks, it is shown that although increasing temperature can reduce looping by encouraging exploration, this method is not the one that can repair the underlying probability mismatch. Thus, although looping is avoided, the responses are still longer than optimal. In conclusion, this paper suggests that using temperature adjustment as a stopgap, while further methods are required to solve the underlying problem.

**Compliance With Llm Reviewing Policy:**

Affirmed.

**Final Justification:**

My concerns are solved and I would like to raise my score.

**Key Questions For Authors:**

Q1: Row37 says generations under high temperature are still longer than necessary. Could you explain how you define the "necessary length" if there is any metric or human label included?

Q2: In this paper, temperatures from 0.0 to 1.0 are tested, and shown that there is less looping under higher temperatures. I am just curious about the result under temperatures higher than 1.0. Usually, I think a very high temperature will lead to a bad reasoning chain, even make the content hard to understand, but when will this happen? Is there a trade-off between less looping and higher accuracy?

**Limitations:**

Please see the weaknesses part above, thanks.

**Strengths And Weaknesses:**

## Strengths
1. I think this paper is well written with sufficient case studies. The experiment part covers a wide range of reasoning LLMs from different families with different sizes.

2. This paper states some interesting findings, such as student models tend to loop more than teachers, and that loopings are caused by risk-aversion and an inductive bias toward temporally correlated errors.

3. The use of graph reasoning tasks is an ingenious design. The looping mechanism is well explained under this synthetic graph reasoning task.



## Weaknesses
1. This paper states that *errors in learning* is the key cause of looping. However, there is no systematic analysis about the difference between the training distributions of teacher and student models.

2. Although this paper demonstrates the looping mechanism with a graph reasoning task. I think it is just a toy task and differs from real-world reasoning tasks. If this paper can formalize the tasks in AIME or GPQA into this kind of graph reasoning task, and provide some analysis about risk-aversion and bias for temporally correlated errors, it will be more convincing.

3. In section 1.1, this paper mentions some other approaches that can mitigate looping, such as [1], [2] and [3]. I think these methods can be considered as baselines, but none of them are included in the experiments. This paper states that increasing temperature can be a stopgap to mitigate looping. Is this method better than these baselines in addressing the looping problem? If not, this paper also finds many underlying reasons for looping. Based on these identified reasons, is it possible to increase the accuracy of reasoning tasks with these baseline methods? The lack of these analyses makes the overall contribution feel limited. It seems that this paper only identifies problems, but does not provide a solution.


## Minor suggestions, not weaknesses
Section 2.1 is a bit hard to read. Could different parts of each example (model/ dataset/ task/ model response/ reason for looping) be highlighted with different colors or formatted into a table?


[1] Su, Yixuan, et al. "A contrastive framework for neural text generation." Advances in Neural Information Processing Systems 35 (2022): 21548-21561.

[2] Welleck, Sean, et al. "Neural text generation with unlikelihood training." arXiv preprint arXiv:1908.04319 (2019).

[3] Li, Huayang, et al. "Repetition in repetition out: Towards understanding neural text degeneration from the data perspective." Advances in Neural Information Processing Systems 36 (2023): 72888-72903.

---

> ### Author Rebuttal · Authors · 2026-03-30
>
> Thank you for thoroughly reading our paper and providing constructive feedback. We feel encouraged that you found our findings interesting and the graph reasoning task design
> ingenious. Below, we respond to your questions.
>
> # Controlling for Training Distribution (W1)
>
> By restricting our comparison to the same model family, we can control for the training distribution. For e.g., within the OpenThinker3 family, the 7B and 1.5B models are trained on the exact same openly available OpenThoughts-3 dataset, yet the 1.5B loops significantly more than the 7B, demonstrating that errors in learning driven by capacity are a key cause (note: if there were no errors in learning, both models would have the same learned distribution, and we would not expect different looping rates). Moreover, this is also why we introduce the synthetic setting, which lets us control such factors very precisely.
>
> # Synthetic-to-Real Gap (W2)
>
> Please see our response to Reviewer tXgn, where we discuss the evidence we do provide and put our contribution in perspective.
>
> # Alternate Methods (W3)
>
> **Why we focus on temperature** We focus on temperature because: 1. It is essentially the only method modern open reasoning model providers use to mitigate looping; 2. The other methods ([1, 2, 3]) discussed in our related work are from the GPT-2 era and, to the best of our knowledge, none are used in modern reasoning model training pipelines. These are not simple baselines one can plug in; evaluating them would require retraining reasoning models with significant compute, and making even one work in the modern context would likely require a separate paper-level effort.
>
> **Can these methods mitigate shortcomings of temperature?** At a high level, all three methods advocate penalizing repetition either at training time or inference time. We did try using a repetition penalty at inference time with OpenThinker models and found that it made the model unstable, producing gibberish. Among training-time interventions, our margin-based experiments (Sec 4) are similar in spirit to [3] (reducing repetition in training data), and we find they help to some extent. Based on this, processing training data to reduce repetition may be a worthwhile future direction (although this requires care, as naively removing steps like backtracking can hurt accuracy).
>
> More importantly, we do not think these methods help in a fundamentally different way than temperature, which is a stopgap. Temperature is reasonably effective at reducing looping, but is a stopgap in that it still leaves training errors lurking, showing up e.g. in response lengths being longer than necessary. These methods are not designed to mitigate such errors (e.g., they do not decrease hardness of learning), and newer approaches as discussed in Sec 5 are needed.
>
> **On not providing solutions.** A first step toward effective mitigations is understanding the forces at play and how existing mitigations fall short, which was the main focus of our paper. This already required extensive experiments across many open models and synthetic settings. We hope this lays a foundation for future work on more effective mitigations.
>
> Your question made us realize that the above points may not be obvious from our current presentation, and we will discuss this more in our revised version. Thank you!
>
> # Longer Than Necessary Generations (Q1)
>
> By "necessary," we refer to two concrete comparisons. In the real-world experiments, larger/teacher models produce substantially shorter responses than smaller/student models at the same temperature (e.g., Fig 1), suggesting the extra length in student responses is not necessary for solving the problem. In the synthetic experiments, we compare against the perfect learner: on G(5,5) with exploration at temp 1, the trained model has average response length 105.5 vs. 24.8 for the perfect learner, a roughly 4× blowup (Sec 3.3); in the correlated-errors setting, ~1.5× (Sec 4.1). Thus, even when temperature removes surface-level looping, the trained model still takes many more steps than a model that has correctly learned the distribution.
>
> # Temperature Range (Q2)
>
> We tested temperatures only up to 1 because: (i) temp. 1 is already higher than what most open model providers recommend for peak accuracy (usually 0.6-0.8), and (ii) looping already reaches 0 for most models by temp. 1. Your intuition is right: in the limit, high temperature leads to a uniform distribution with 0 looping but also 0 accuracy. The temperature achieving highest accuracy is not always the one with lowest looping. For example, for Phi-4 reasoning (Fig 8), accuracy peaks at temp. 0.8 while looping is still decreasing. Importantly, we think this tradeoff is a property of temperature, not inherent: our risk-aversion mechanism shows that if the underlying errors in learning were addressed, the model would assign more probability to progress-making actions, decreasing looping and increasing accuracy at the same time.

---

> > ### Author Rebuttal · Reviewer_cMjw · 2026-04-03
> >
> > Thanks for your detailed reply. Most of my concerns are addressed, but I have a bit follow up questions.
> >
> > Follow-up questions for W1: Does this mean that this method is only limited to LLMs that have published their training data?
> > Follow-up questions for W2: It would be valuable to additionally include results on real data instead of just claiming about it.

---

> > > ### Author Response · Authors · 2026-04-05
> > >
> > > Thank you for engaging with our responses and for the follow-up questions. We are glad that most of your concerns are addressed. Below we respond to the remaining questions.
> > >
> > > # Follow-up on W1
> > >
> > > Our hypothesis (errors in learning as a key cause of looping) does not rely on training data being publicly available. We do not analyze the training data itself. What we need is just to know that models were trained on the same distribution, which is documented in the respective papers. This holds for both OpenThinker3 (1.5B/7B trained on the same OpenThoughts-3 data) and the Qwen family (DeepSeek-R1 distilled 1.5B/7B/32B, all trained on the same 800k samples as stated in the DeepSeek-R1 paper). The point is: if two models trained on the same data loop differently, the difference must come from different errors in learning (which are caused due to capacity in this case), not from training data differences. This is what we observe across both families.
> > >
> > > # Follow-up on W2
> > >
> > > Thank you for this suggestion. Below, we summarize our results on real models and further clarify the mapping between the synthetic and real settings.
> > >
> > > **Results on real models.** In Section 2, we evaluate looping across several open reasoning models on AIME and GPQA. In Example 1 (Appendix B.1), we go further: we identify the exact point where the student (OpenThinker3-1.5B) starts looping, and run both student and teacher (QwQ-32B) from the same prefix. The teacher produces key progress-making insights in 34/50 high-temperature continuations vs 1/50 for the student. This shows that the student's looping point exactly coincides with a point where the student finds it hard to produce the progress-making action, directly in line with the risk-aversion mechanism. We discuss this and the challenges in making the mapping more precise in our response to Reviewer SsUF. We will add a more prominent discussion of these results in the revised version.
> > >
> > > **Further clarifying the mapping between graph-reasoning task and real traces.** In the graph reasoning task, the model starts at the source node and does a random walk with the aim of reaching the goal node. At any node in the graph, it takes an action which is simply the choice of next node to pick. A cyclic action is an action that takes the walk to an already visited node, and a progress-making action is an action that moves the walk closer to the goal node.
> > >
> > > The analogous notions for reasoning language models can be defined at the level of short spans of tokens ("chunks"). A chunk is a span of tokens that implements a single logical step in the chain-of-thought (for example, a line or small paragraph). Conditioned on the prefix generated so far, a language model induces a distribution over possible next chunks; these are the actions in our abstraction. So a reasoning language model can be thought of as doing a random walk, where at each step it conditions on the chunks generated so far, and samples a new chunk. A cyclic action corresponds to a chunk that repeats something already done before (e.g., restating a previously stated fact), while a progress-making action corresponds to a chunk that moves closer to the final goal (e.g., taking the next step in the proof or narrowing down the candidate set). The high-degree root node in the star graph can be thought of as any point in a CoT where the model must choose among multiple plausible continuations. This can happen at multiple points in a trace, so the star graph represents a local structure within a longer CoT (we discuss this locality also in our response to Reviewer SsUF). We also discuss in Appendix C how this mapping relates to Proposition 1.
> > >
> > > We also want to emphasize that this mapping does not capture all the complexities of a real reasoning model trace. And that is by design: our goal with the graph reasoning task is not to capture the full complexity of real reasoning traces, but to have a minimal setting where we can isolate and understand the forces behind looping.
> > >
> > > We hope this helps answer your questions. Your feedback has helped us improve the paper, and we will make sure to discuss these points more prominently in the revised version.

---

### Official Review · Reviewer_tXgn · 2026-03-12

**Soundness:** 3
**Presentation:** 3
**Significance:** 3
**Originality:** 2
**Overall Recommendation:** 4
**Confidence:** 3

**Summary:**

This paper studies looping in reasoning models, especially at low temperature or greedy decoding, and argues that a key cause is "errors in learning," meaning mismatch between the training distribution and the learned model. The empirical part shows that looping is more common at low temperature, tends to be worse in smaller models, and can be substantially worse in distilled students than in their teachers. The paper then introduces a synthetic graph-reasoning setup and uses it to illustrate two mechanisms: hardness-induced risk aversion and temporally correlated errors. The main claim is that higher temperature reduces looping mainly by encouraging exploration, but does not fix the underlying learning problem, so high-temperature generations can still be unnecessarily long.

**Compliance With Llm Reviewing Policy:**

Affirmed.

**Final Justification:**

Overall, the rebuttal successfully addresses my main concerns, and I am therefore raising my score.

**Key Questions For Authors:**

- Can the authors provide a direct probe on real reasoning models that distinguishes hardness-induced risk aversion from temporally correlated errors, rather than relying mainly on qualitative trace analogies?

- How sensitive are the main open-model conclusions to the operational definition of looping, especially across semantically repetitive but not exactly repeated traces?

**Limitations:**

The paper is careful to note that the synthetic setup is intentionally simplified and that other forces may contribute in real language models. That caveat is appropriate, but it also highlights the main limitation: the evidence for mechanism in real models remains indirect.

**Strengths And Weaknesses:**

**Strengths**

- The paper focuses on a real and important failure mode of reasoning models, and the open-model study is directionally convincing. The student-teacher gap in looping is a particularly interesting observation.

- The synthetic graph setup is well chosen for isolating mechanisms. It gives the paper a clearer explanatory structure than a purely empirical study on black-box model traces would have.

- The distinction between hardness-based risk aversion and temporally correlated errors is conceptually useful.

**Weaknesses**

- The central causal claim is broader than the direct evidence. The open-model results are correlational, while the mechanistic support comes from a synthetic setup that is far removed from actual language-model training and decoding.

- The claim that student-teacher looping gaps point to "errors in learning" is plausible but under-identified. Distillation mismatch, decoding artifacts, data curation differences, RL post-training effects, and prompt-format sensitivity could also contribute, and the paper does not cleanly disentangle these possibilities in the real-model experiments.

- The synthetic experiments are elegant but narrow. It is not yet clear how much the specific reset-based graph dynamics, especially the explicit cyclic reset action, transfer to natural language reasoning traces.

---

> ### Author Rebuttal · Authors · 2026-03-30
>
> Thank you for the thoughtful review. We are glad the student-teacher looping gap resonated with you, and that you found the synthetic setup well chosen for isolating mechanisms. Below, we respond to your questions.
>
> # Synthetic-to-Real Gap (W1)
>
> We discuss the evidence we do  provide, challenges in going further and put our contribution in perspective.
>
> **Evidence we provide**. In Example 1 (Appendix B.1), we take a looping trace from the student (OpenThinker3-1.5B), identify the prefix just before the final loop, and run both student and teacher (QwQ-32B) from this prefix. With greedy decoding, the teacher makes substantial progress while all student continuations loop. To quantify probability mass on progress-making actions, we sample 50 continuations at temperature 0.7 and use GPT-5 to judge whether each contains a key structural insight: teacher 34/50, student 1/50. This is consistent with the risk-aversion mechanism: the teacher puts significantly more mass on progress-making actions, and it not looping under greedy decoding suggests relatively less preference for the easy cyclic action. We conduct a similar analysis for the second mechanism (Example 2).
>
> **Challenges in going further.** While this example is consistent with the proposed mechanism, it admittedly does not capture all its details. For instance, we show that the teacher puts significantly more mass on progress-making actions, but we do not precisely characterize all possible actions and the mass on each. Such precision is challenging for two reasons. First, actions in real models live in semantic space: detecting whether a specific insight occurred is feasible (a binary LLM-as-judge question), but clustering all possible continuations into discrete actions with reliable probability estimates is not, due to the difficulty of mapping long token-level outputs to semantic categories and the high-dimensional nature of the outputs. Second, multiple forces act simultaneously and confound each other's effects. Even within risk-aversion, our synthetic experiments reveal two sub-forces (probability diffusion across indistinguishable alternatives, and overestimation of the easy cyclic action), both consistent with the progress-making action receiving low mass but having different preferences for distributing the overall mass, something we cannot disentangle in a real model (see Appendix E and F for further examples of forces interacting).
>
> Given these challenges, while fully understanding looping in real models is desirable, we think it is a longer-term community project. As a step towards this, we focus on disentangling the possible forces at play in the simplest setting possible. While this may seem like a lower bar, prior to our work we are not aware of any work that achieves even this for looping. This way of making progress on understanding complex systems using controlled settings is standard scientific methodology, and many influential works in the community have made a similar contribution (e.g., [Bachmann et al.](https://arxiv.org/abs/2403.06963) ).
>
> Upon reading your review and others, we realize these points are not sufficiently emphasized in our presentation and will fix this in the revised version (thank you!).
>
> # Distinguishing the Mechanisms (Q1)
>
> The two mechanisms produce distinct looping patterns in real traces. In risk-aversion (Example 1), the model is hesitant to make progress and defaults to restating known facts. In correlated errors (Example 2), the model does make progress but makes a mistake, hits a contradiction, restarts, and repeats the same mistake. At the level of these looping patterns, the two mechanisms are distinguishable. At a deeper level, the underlying forces can also interact in real models (e.g., Appendix E shows correlated-errors looping driven by a combination of hardness and inductive bias), making it challenging to fully disentangle them without training-time interventions. This is precisely why we study the synthetic setting, where it is easier to disentangle the forces at play.
>
> # Errors in Learning as a Cause (W2)
>
> By restricting our comparison to the OpenThinker3 family, we control for these factors: the 7B and 1.5B models share the exact same dataset, no RL, same decoding, and same prompt format. Yet the 1.5B loops significantly more, strongly suggesting that errors in learning (from limited capacity in this case) are a key cause.
>
>
> # Reset-Based Graph Dynamics (W3)
>
> Please see our response to Reviewer SsUF.
>
> # Looping Definition (Q2)
>
> This is a great question. In Appendix A.3, we ablate across various values of n and k and show that broad trends are consistent. Regarding semantic looping, our qualitative impression is that models which syntactically loop more also tend to exhibit more semantic repetition (also visible in response length measurements, a coarse proxy). Capturing semantic looping more precisely is challenging (it is not even well-defined) and is a worthwhile direction for future work.

---

> > ### Author Rebuttal · Reviewer_tXgn · 2026-04-04
> >
> > Thanks for your detailed reply. My main concerns have been addressed. I will raise my score to 4.

---

> > > ### Author Response · Authors · 2026-04-05
> > >
> > > Thank you for engaging with our responses and for your willingness to raise the score! We really appreciate it. We noticed the updated score may not have been reflected yet in the system. Would you mind updating it when you get a chance? Thanks again for your thoughtful feedback.

---

### Official Review · Reviewer_V8Ug · 2026-03-13

**Soundness:** 3
**Presentation:** 3
**Significance:** 3
**Originality:** 3
**Overall Recommendation:** 5
**Confidence:** 4

**Summary:**

This paper investigates why reasoning models tend to produce repetitive loops under greedy or low-temperature decoding. Empirically, the authors show that looping is common in open reasoning models (more frequent in smaller models and in distilled students relative to their teachers), and tends to occur more often on harder problems. To explain these observations, the paper introduces a synthetic graph-reasoning setting and studies two mechanisms: (1) Hardness-induced risk aversion, where probability mass over a hard progress-making action is diffused across many alternatives. This makes an easy cyclic action comparatively more attractive (2) Temporally correlated errors, whereby small local errors persist over time and repeatedly favor the same choices. The paper further argues that increasing temperature mainly reduces looping through exploration, without addressing the underlying learning errors.

**Compliance With Llm Reviewing Policy:**

Affirmed.

**Final Justification:**

The rebuttal addresses most of my concerns. Some remain unresolved, but I think they might be left for future work. Overall speaking, I'm leaning towards acceptance since this is an interesting paper.

**Key Questions For Authors:**

1. The paper argues that maximum-likelihood training can diffuse probability mass across indistinguishable options, making an easy cyclic action comparatively more attractive. Can the authors test this claim directly on real reasoning models, for example by analyzing token- or step-level probability distributions around looping prefixes and comparing them against non-looping continuations?

2. The synthetic analysis suggests that increasing the effective margin between revisiting a previously explored bad action and taking a progress-making action may reduce looping. Are there realistic analogues of this idea for real LLMs, such as training-data modifications, preference shaping, or test-time penalties?

3. Which qualitative predictions of the synthetic theory do the authors expect to transfer most reliably to pretrained LLMs?

**Limitations:**

Yes

**Strengths And Weaknesses:**

### Strengths

1. Presentation: The paper is clearly written and easy to follow. The two motivating examples are well chosen and connect to the later analysis.

2. Soundness: The empirical observations on open reasoning models are interesting and well motivated. The findings that looping is more severe at low temperatures, in smaller models, and in distilled students than in their teachers provide a useful starting point for the paper’s central claims.

3. Originality: The synthetic graph-reasoning framework is a novel abstraction of the phenomenon. It provides a clean setting in which the proposed mechanisms can be studied in a controlled way.

4. Significance: The synthetic experiments are insightful. In particular, they make the roles of indistinguishability, exploration, and temporally correlated errors clearer.

### Weaknesses

1. Significance: One limitation is that the paper mainly identifies and explains the failure mode, but does not yet provide a concrete solution. This is not necessarily a fatal issue for a paper focused on diagnosis, and I would not penalize it. However, the practical impact would be stronger if the paper included at least one implemented mitigation that reduces looping.

2. Soundness: The concern is whether the synthetic setup, especially training Transformers from scratch, faithfully reflects the behavior of real LLMs. Pretrained reasoning models inherit strong priors from large-scale pretraining, and these priors may substantially alter the learning dynamics underlying looping. For example, for the claim that **maximum-likelihood training diffuses probability mass across indistinguishable options**, it would strengthen the paper to test this more directly on real LLMs and real responses.

3. Significance: The synthetic experiments currently serve mainly as explanations, but less for prediction or intervention. A stronger mechanistic explanation would ideally support predictions in unseen settings and suggest interventions that can be verified empirically. For example, for the margin-based analysis, the paper did not show whether analogous interventions (during training or at inference time) can reduce looping in real reasoning models.

4. Presentation (minor): The motivating examples are effective, but they would likely be easier to read if accompanied by more intuitive figures or schematic illustrations in the main text.

5. Soundness: The open-model experiments show that looping exists and correlates with factors such as model size, but the synthetic experiment is not able to fully explain this factor.

---

> ### Author Rebuttal · Authors · 2026-03-30
>
> Thank you for the thoughtful and constructive review. We feel encouraged that you found our synthetic framework a novel and insightful abstraction. Below, we respond to your questions.
>
>
> # Synthetic-to-Real Gap (W2, W3, Q1)
>
>
> Please see our response to Reviewer tXgn for this, where we discuss the evidence we do provide, explain why going significantly further is technically challenging, and put our contribution in perspective.
>
>
> # Realistic Analogues of Margin-Based Interventions (W3, Q2)
>
>
> Thank you for this question. The margin experiments in Section 4 already demonstrate that modifying training traces to discourage revisiting past actions reduces looping. For real models, analogous interventions can operate at inference time or training time. At inference time, a natural analogue is a repetition penalty. We ran preliminary experiments with OpenThinker models and found that the model frequently degenerates into gibberish, both because it was not trained with such a penalty, and because the penalty operates at the token level rather than the semantic level (e.g., one should not penalize "the" just because it appeared before). Both issues can be addressed through training-time interventions: RL training where rollouts use a repetition penalty can stabilize the model, and for semantic-level intervention, one can pre-process teacher traces to reduce semantic repetition before distillation, directly analogous to our margin-based training in Section 4. Exploring these training-time interventions on real reasoning models would require significant compute and is beyond the scope of this paper. It is a great direction for future work though, building on the understanding we develop!
>
>
> # Synthetic Experiments and Model Size (W5)
>
>
> Our theory (Proposition 3.1) predicts looping whenever a hard action coexists with an easy cyclic action. Limited model capacity is one natural source of such hardness: a smaller model is more likely to collapse distinct contexts, making progress-making actions harder to distinguish.
>
>
> # On Implementing Mitigations (W1)
>
>
> A first step toward effective mitigations is understanding the forces at play and how existing mitigations (such as temperature) fall short, which was the main focus of our paper. This already required an extensive set of experiments (looping evaluations across many open models and extensive synthetic experiments). We hope this lays a foundation for future work on both deeper understanding and more effective mitigations.
>
>
> # Qualitative Predictions That Transfer to Real Models (Q3)
>
> Thank you for the great constructive question! We list some of our main qualitative takeaways that we expect to transfer:
>
> 1. **Errors in learning as a major cause of looping.** We already see this in our open-model results, where students loop far more than their teachers. If the student had perfectly learned the teacher distribution, this gap would not exist.
>
> 2. **Hardness-based risk-aversion.** We believe one force at play is the student putting relatively less mass on hard progress-making actions compared to easy cyclic actions. We show this in a real trace in Example 1 (as also discussed in rev tXgn response), where the student’s looping point is considerably harder for the student than the teacher when evaluated at high temperatures.
>
> 3. **Repeatedly making the same mistake.** This is another looping pattern we clearly see in Example 2, where the model makes a mistake, reaches a contradiction, restarts, and falls into the same error. Multiple forces can contribute to this; in the main text we discuss how an inductive bias for temporally correlated errors can cause it, and in Appendix E we show how hardness can also contribute.
>
> 4. **Temperature as a stopgap.** In our synthetic experiments, temperature cannot fix underlying learning errors, and even at high temperature the model produces longer responses than a perfect learner. We see similar trends in our open-model experiments. Moreover, when we qualitatively inspect traces, student models tend to waste many tokens (either repeating already stated facts or exploring low-potential directions), consistent with higher errors that temperature does not fix.
>
> 5. **A catalyst for looping.** In real model traces, the model becomes more confident as it loops, making it harder to escape (Figure 15). In Appendix F, we reproduce this exact trend in our synthetic setting, where the key insight is that even a very tiny fraction of repetitive traces during training is sufficient to cause this self-reinforcing behavior. Pretraining on large-scale data likely contains instances of copying behavior that could play a similar role.

---

> > ### Author Rebuttal · Reviewer_V8Ug · 2026-04-02
> >
> > I would like to thank the authors for their response. Some of my concerns are not fully resolved, such as the gap between synthetic setting and real LLM, but I think this might need further efforts from the community. The paper itself is fine as an early work that tries to formalize looping behavior in reasoning models, so I will keep my positive score.
> > Regarding the claim about model size, I hope the author could provide more detailed discussion, formalization, or experiment. I will further raise my score if this point is addressed properly.

---

> > > ### Author Response · Authors · 2026-04-02
> > >
> > > We really appreciate your thoughtful engagement with our work and your openness to our response. Your willingness to engage iteratively and suggest concrete directions has genuinely helped us strengthen the paper, and we are grateful for that.
> > >
> > > **Additional experiment:** To demonstrate the effect of model size on looping in our synthetic setup, we ran an additional experiment. For the G(5,3) graph (hardness at root; section 3), we train models of three different sizes: 85M (original), 14.2M, and 7.1M parameters, keeping all other hyperparameters identical. The larger model barely loops and achieves close to perfect accuracy, while the smaller models loop significantly at temperature 0 (the expected root-source loop), with looping decreasing with temperature. We show results for temperature 0 and 1 below.
> > >
> > > **Temperature 0**
> > >
> > > | Model | Accuracy | Looping Count | Response Length |
> > > |:-----:|:--------:|:-------------:|:--------------:|
> > > | 7.1M | 0.075% | 72.6 | 154.1 |
> > > | 14.2M | 15.3% | 16.7 | 38.1 |
> > > | 85.0M (original) | 99.2% | 4.6 | 20.8 |
> > >
> > > **Temperature 1**
> > >
> > > | Model | Accuracy | Looping Count | Response Length |
> > > |:-----:|:--------:|:-------------:|:--------------:|
> > > | 7.1M | 3.8% | 4.8 | 24.7 |
> > > | 14.2M | 19.6% | 3.4 | 22.9 |
> > > | 85.0M (original) | 98.8% | 3.3 | 22.6 |
> > >
> > > We chose G(5,3) for this ablation deliberately. For G(5,5), the source of hardness is optimization: even a large model cannot learn the correct first edge from the root, as the problem reduces to learning a hard parity instance (see Hu et al. and Bachmann et al.). Model capacity is not the bottleneck there. With G(5,3), however, the optimization hardness is no longer present, as the 85M model can solve the task near-perfectly. This makes it the right setting to isolate model capacity as the source of hardness: we expected that shrinking the model would make the root decision harder to learn, and this is exactly what we observe.
> > >
> > >  More generally, by Proposition 3.1, whenever a hard action coexists with an easy cyclic action, the hard action distributes its probability mass over indistinguishable alternatives, while the easy action retains its mass, leading to looping under greedy decoding. In the experiment above, the source of this hardness is model capacity: the smaller models lack the depth to resolve which outgoing edge from the root leads to the goal, so probability mass gets diffused across the alternatives. We also verified this directly by examining the root-node transition probabilities, which are indeed more diffused for the smaller models.
> > >
> > > We will add this experiment to the final version of the paper. Thank you!

---

### Official Review · Reviewer_SsUF · 2026-03-13

**Soundness:** 2
**Presentation:** 4
**Significance:** 3
**Originality:** 3
**Overall Recommendation:** 4
**Confidence:** 4

**Summary:**

The paper sets out to study looping, the phenomena where chain of thought / "reasoning" traces of modern language models backtrack (often surfaced in the form of "wait wait") many times, sometimes repeating the same text verbatim. The language model analysis proceeds by counting the percentage of AIME rollouts containing at least 20 repetitions of the same 30-grams. A list of observations are made based on this, about the effect of temperature, model size, and problem hardness on the occurence of looping. Two hypotheses are proposed with examples from model reasoning traces. From here, the paper tries to show the plausibility of these hypotheses and observations by setting up a toy graph formulation, akin to empirical theory. It is shown that the same observations occur in this constructed graph formulation. The loop is NOT closed by studying whether the hypotheses also occur in language models.

**Compliance With Llm Reviewing Policy:**

Affirmed.

**Final Justification:**

I think this work contributes an interesting potential intuition for the problem of looping observed in reasoning language models. However, there is limited evidence on their proposed hypotheses applying to actual reasoning models. Instead, the paper goes deeper into the conceptual framework through synthetic examples. I think people interested in looping could learn some interesting ideas from reading this paper, so I am leaning toward accept even though I think the paper effectively overclaims in the title/abstract for how far it goes in explaining these phenomena on actual reasoning models and whether their proposed hypotheses are superior to alternatives.

**Key Questions For Authors:**

Is there any causal explanation the authors have for why higher capacity models loop less? This conflicts with Sinha et al. (2025) mentioned above, and while the phenomena being studied have important differences (they study increase in error rates, not looping), I would have expected a similar trend to hold here. So maybe there is some interesting causal phenomena why these two results differ, and I wonder if the authors have any insights or intuition that could help me grok this.

 Why do you think that temporal correlation in errors is a property of the transformer architecture, and not say the next token prediction objective?

Is there any evidence on the central claim: that language models end up dividing their probability over progress making actions leading to looping and temporal correlation in errors?

**Limitations:**

At the risk of being too candid, I feel a bit let down by the paper. The story is setup really well. The experiments are interesting. However, I felt a fundamental disconnect between the motivation on language models and the analysis on the synthetic graph setting, that was never really connected for me. This limitation is not sufficiently highlighted in the paper.

**Strengths And Weaknesses:**

### Strengths

- The paper sets out to study a targeted, well-scoped out, and interesting problem: looping in thinking traces of a language model. It dives deep into this, building the reader's intuition along the way with interesting examples (highly appreciated!) and logical experiment flow.

- The synthetic graph reasoning task is a good fit for the study! I appreciate the experimental design to illustrate the hypothesised problem of indistinguishable hard actions.

- I think comparing distillation to RL trained reasoning models is an interesting experiment.

### Weaknesses
Unfortunately, I think the flow doesn't connect at the end from the synthetic setting to claims about language models. My main concerns are about the generalizability of the observations from the synthetic settings used for the proposed hypotheses and intuitions, to real-world language model behaviour. If the rebuttal convinces me about this, I am quite willing to increase the score to acceptance. At the end of reading the paper, I don't think I learnt anything convincing about why reasoning language models loop via "wait wait wait", just an interesting conjecture.

- It is hypothesised that probability getting distributed across indistinguishable actions is the reason behind why models choose easy looping actions over hard progress actions. I appreciate the demonstration in the synthetic setting. I think this is a plausible hypothesis for what might be happening in real language models, but the bridge to whether this is really what happens in real language models is not demonstrated in this paper, which is a crucial burden of proof to motivate the experiments in the toy setting.

- "the random-walk variant can be seen as a model for reasoning language models that explore multiple strategies, backtrack, and restart." I disagree with this claim. The random walk model resets to the start node with a probability 0.3. I think this is too extreme to model real language model behaviour. First, they don't reset to the initial node necessarily. Second, the probability of looping intuitively feels a lot lower. Its not discussed how these discrepancies would affect any interpretations that can be made about these results that would apply to practical settings. For example, one clear issue with this setup is that at temperature 1 (normal sampling), the probability of reaching the goal would be extremely low for the random walks even at moderate lengths (e.g. 10) as it would very often get reset before reaching the goal.

- In multiple places it is hypothesised that hardness due to smaller capacity makes small models loop more, perhaps because more of the context collapses making progress actions harder to distinguish. I did not see any real evidence for this explanation in the paper. The toy setting constructed illustrates this is possible, but is very far from showing this is what happens in language models. I am especially concerned due to the competing hypothesis from prior work (Sinha et al. 2025) that larger models actually compound in errors faster, with their explanation being that they are stronger "next token prediction" simulators shown with causal manipulation of the trajectories. Under their hypothesis, in this paper's setting, larger models might simply be looping less because they make fewer "mistakes" or not-progress-making actions in the first place, leading to lower chances of compounding errors due to a much stronger initial state of the dynamical system. I would love to see some experiments on language models in natural tasks to shed more light on what is actually going on here, as this is one of the central claims of the paper.

- The paper's analysis on actual language models is mostly based on a definition of looping as 30-grams appearing atleast 20 times in the reasoning trace. I recognize these parameters were set conservatively to be sure that the behaviour occurring is highly likely to be looping. I also appreciate that Appendix A.3 shows that looping percentages are not much higher for 20 ngrams appearing 20 times. But even that seems too strict. How would the main results vary if you looked for softer definitions of looping, say a 10-gram appearing 10 times. Does this really add a lot of false positives? I would be surprised. But this also would reduce a lot of false negatives in my opinion. I guess the main point is this is hard to know, could change the results, and I would have really liked an analysis on what false positive / false negative tradeoff is struck at the current and different hyperparameter choices.

- Similarly, with the temperature analysis, the bridge from theory to practice is quite a surface level correlation. The theory developed suggests that probability division across indistinguishable actions would reduce their probability, and then temperatue only acts as a stop-gap by promoting exploration. This is a fine hypothesis, but I don't see any analysis on whether language models really have the problem of dividing probability too much on progress-making actions. The only evidence in this context on language models presented is that smaller models produce longer chains, and students have lower accuracy than the teacher. I don't think the explanation loop is really closed here, and is far from convincing.

- It is claimed in Section 4 that transformers have a bias for temporally correlated errors. It is unclear why this bias is transformer specific, and not applicable to any architecture trained with next token prediction. No non-transformer models are tested, and its not even clear what part of the transformer architecture (self attention? but why?) would lead to this bias.

### Suggestions

Where you mentioned looping has been observed since the early days of large language models, it might actually be interesting to discuss even earlier work like Sequence Level Training with Recurrent Neural Networks who discuss a very similar problem to this paper back in 2015 (though back then the language models weren't large by today's standards). And it might also be useful to the reader to discuss how a variant of this effect, self conditioning as recently studied in Sinha et al. 2025, The Illusion of Diminishing Returns: Measuring Long Horizon Execution in LLMs, relates to looping. It seems like the section on "bias for temporally correlated errors" is pretty similar to the self-conditioning effect discussed in Sinha et al. who explain this using next token predictor behaviour learnt by a model.

---

> ### Author Rebuttal · Authors · 2026-03-30
>
> Thank you for the detailed review. We appreciate that you find the problem well-scoped and the synthetic setting a good fit. Below, we respond to your questions.
> # Synthetic-to-Real Gap
> Please see our response to Rev tXgn.
> # Role of Self-Conditioning
> This is a great point! We agree that self-conditioning (conditioning on an erroneous trace making looping more likely) is an important force. We view it as an amplifier that works in conjunction with other looping mechanisms, which need to initiate the loop first. We already illustrate this at various points:
>
> **Catalyst force (App F).** We demonstrate in our synthetic setting how self-conditioning amplifies proposed mechanisms: once a model has looped, conditioning makes it increasingly confident about continuing, explaining the same confidence buildup we observe in real traces.
>
> **Varying prominence.** We see examples where self-conditioning plays a big role and others where it seems benign. In Ex. 1 (App B.1), the student prefix just before looping is relatively benign and the teacher does not loop from it, suggesting risk-aversion rather than conditioning is the main driver. In Ex. 2 (B.2), the teacher escapes from an early prefix but gets stuck from a later one containing extensive semantic looping, suggesting conditioning plays a role. Thus, conditioning is not always prominent, and even when it is, it acts on top of looping initiated by other mechanisms.
>
> Overall, self-conditioning is an important amplifier worth understanding more (we highlight this in future directions in sec 5). From your review, we realize these points may have become buried (thank you!); we will surface them in the revision along with discussing Sinha et al.
> # Sinha et al.
> We don’t see a conflict with Sinha et al. As you note, larger models make fewer errors, generating cleaner traces with less to condition on; thus it also works in the direction of decreasing looping with model size. Their finding that "scaling sometimes leads to stronger conditioning" is about a controlled counterfactual where models see the same injected erroneous history; in natural generation, the larger model produces its own (less erroneous) history. Additionally, they find reasoning models (our focus) are less affected by self-conditioning.
> # Modeling choices
> The reset-to-start shouldn't be viewed as resetting to CoT  beginning. The start and root can be any local point within a CoT where the model faces a hard decision; the "reset" models any cyclic action that returns to a nearby decision point. We chose this just as the simplest instantiation and Prop. 3.1 is agnostic to the form of the action.
>
> On prob 0.3, the specific value isn't essential. By Prop. 3.1, for any p, a sufficiently large n produces looping (e.g., p=0.05 needs n≥20), and in real reasoning the number of plausible continuations can easily be large. So 0.3 is just a convenient choice for modest-size graphs trainable with reasonable compute; one could set it much lower with larger graphs. Moreover, self-conditioning can increase the cyclic action probability over time even if it starts small (App F).
>
> Temp-1 sampling: We allow generation up to a maximum length sufficient for the model to reach the goal and the perfect learner achieves 100% accuracy at temp 1.
> # Looping Definition
> We present looping % at temp 0 for the 10-gram setting with different thresholds k. With (n=10, k=10), absolute looping increases significantly. Manual inspection of responses flagged by (10,10) but not (30,20) showed nearly all were false positives (e.g., benign reuse of expression `(33/(2√14)) y`). Since ground truth is unavailable, formal FP/FN analysis is infeasible. That said, the relative trend of looping increasing with decreasing model size remains consistent.
>
> |Model|10,10|10,20|10,40|30,20 (orig.)|
> |---|---|---|---|---|
> |Qwen 1.5B|89|80|77|76|
> |Qwen 7B|74|58|52|49|
> |Qwen 32B|64|47|41|37|
> |OT 1.5B|84|59|38|30|
> |OT 7B|72|37|16|10|
> |QwQ-32B|49|16|6|4|
>
> # Correlated Errors
> We didn't mean that this bias is exclusive to Transformers (other architectures may exhibit it). What we wanted to emphasize is that it doesn't follow from the NTP objective: the cross-entropy minimizer would assign uniform probability (1/n) to each root child with no temporal correlation. Yet the trained model's deviations from 1/n are correlated across time, indicating an inductive bias of the learned solution. This contrasts with risk-aversion, where the cross-entropy minimizer itself yields the bias.
>
> Relationship to self-conditioning: while these seem similar, temporal correlation does not rely on conditioning. For instance, not only the highest-probability child at the root is correlated across consecutive visits, but also the second-highest (Sec 4). Under greedy decoding the model only visits the top child, so correlation in the second-highest cannot be from conditioning on past choices; it is a property of the learned distribution. We will revise the text to make this clearer.

---

> > ### Author Rebuttal · Reviewer_SsUF · 2026-04-03
> >
> > I appreciate the authors clarifications about my questions and concerns. I have increased my score to reflect this. I would like to see a more transparent, sober statement about what this work achieves in the synthetic setting and what it does / doesn't convincingly tell us about actual reasoning language models that people use, in an early section of the final revision.

---

> > > ### Author Response · Authors · 2026-04-05
> > >
> > > Thank you for increasing your score! We will add a prominent discussion in the final version of the paper about what our synthetic results do and do not tell us about real reasoning models. This should also be helpful for future work that builds on our work.

---

### Decision · Program_Chairs · 2026-04-30

**Decision:**

Accept (spotlight)

**Comment:**

The paper analyzes the looping behavior of reasoning models - the tendency of models to produce repeated patterns in their chain-of-thought. The authors show, through theory and experiments, that the looping behavior is due to errors in learning. They demonstrate through analysis of a graph reasoning problem that language models tend to place higher probability on cyclic actions.

Reviewers appreciated the scope and motivation of the paper, noting that it sets to study an important and well-scoped problem of looping in reasoning models. The reviewers liked the clean synthetic setting of graph reasoning and were satisfied by the analysis and experiments conducted in this setting. Reviewers were initially worried about the toy setting being too restricted, and pointed to an apparent disconnect between this setting and the experimental results with real models. However, it seems that after discussion with the authors, all reviewers agreed that this paper should be accepted.

In light of this, I recommend that this paper is accepted.